# HIF-2α is essential for carotid body development and function

David Macias[1]*, Andrew S Cowburn[1,2], Hortensia Torres-Torrelo[3], Patricia Ortega-Sáenz[3], José López-Barneo[3], Randall S Johnson[1,4]*

[1]Department of Physiology, Development and Neuroscience, University of Cambridge, Cambridge, United Kingdom; [2]Department of Medicine, University of Cambridge, Cambridge, United Kingdom; [3]Instituto de Biomedicina de Sevilla, Seville, Spain; [4]Department of Cell and Molecular Biology, Karolinska Institute, Stockholm, Sweden

**Abstract** Mammalian adaptation to oxygen flux occurs at many levels, from shifts in cellular metabolism to physiological adaptations facilitated by the sympathetic nervous system and carotid body (CB). Interactions between differing forms of adaptive response to hypoxia, including transcriptional responses orchestrated by the Hypoxia Inducible transcription Factors (HIFs), are complex and clearly synergistic. We show here that there is an absolute developmental requirement for HIF-2α, one of the HIF isoforms, for growth and survival of oxygen sensitive glomus cells of the carotid body. The loss of these cells renders mice incapable of ventilatory responses to hypoxia, and this has striking effects on processes as diverse as arterial pressure regulation, exercise performance, and glucose homeostasis. We show that the expansion of the glomus cells is correlated with mTORC1 activation, and is functionally inhibited by rapamycin treatment. These findings demonstrate the central role played by HIF-2α in carotid body development, growth and function.

DOI: https://doi.org/10.7554/eLife.34681.001

*For correspondence:
dm670@cam.ac.uk (DM);
rsj33@cam.ac.uk (RSJ)

**Competing interests:** The authors declare that no competing interests exist.

## Introduction

Detecting and responding to shifts in oxygen availability is essential for animal survival. Responding to changes in oxygenation occurs in cells, tissues, and at the level of the whole organism. Although many of responses to oxygen flux are regulated at the transcriptional level by the hypoxia inducible transcription factor (HIF), at tissue and organismal levels there are even more dimensions to the adaptive process. In vertebrates, the actors in adaptation are found in different peripheral chemoreceptors, including the neuroepithelial cells of the gills in teleosts, and in the carotid body (CB) of mammals. The CB regulates many responses to changes in oxygenation, including alterations in ventilation and heart rates.

The carotid body is located at the carotid artery bifurcation, and contains glomeruli that contain $O_2$-sensitive, neuron-like tyrosine hydroxylase (TH)-positive glomus cells (type I cells). These glomus cells are responsible for a rapid compensatory hypoxic ventilatory response (HVR) when a drop in arterial $O_2$ tension ($PO_2$) occurs (*López-Barneo et al., 2016b*; *Teppema and Dahan, 2010*). In addition to this acute reflex, the CB plays an important role in acclimatization to sustained hypoxia, and is commonly enlarged in people living at high altitude (*Arias-Stella and Valcarcel, 1976*; *Wang and Bisgard, 2002*) and in patients with chronic respiratory diseases (*Heath and Edwards, 1971*; *Heath et al., 1982*). This enlargement is induced through a proliferative response of CB stem cells (type II cells)(*Pardal et al., 2007*; *Platero-Luengo et al., 2014*).

During embryogenesis, the initiating event in CB organogenesis occurs when multipotent neural crest cells migrate toward the dorsal aorta, to give rise to the sympathoadrenal system (*Le Douarin,*

*1986*). Segregation of sympathetic progenitors from the superior cervical ganglion (SCG) then creates the CB parenchyma (*Kameda, 2005*; *Kameda et al., 2008*). This structure is comprised of $O_2$-sensitive glomus cells (type-I cells) (*Pearse et al., 1973*), glia-like sustentacular cells (type-II cells) (*Pardal et al., 2007*) and endothelial cells (*Annese et al., 2017*).

The cellular response to low oxygen is in part modulated by two isoforms of the HIF-α transcription factor, each with differing roles in cellular and organismal response to oxygen flux: HIF-1α and HIF-2α. At the organismal level, mouse models hemizygous for HIFs-α (*Hif1a$^{+/-}$* and *Epas1$^{+/-}$*) or where HIF inactivation was induced in adults showed contrasting effects on CB function, without in either case affecting CB development (*Hodson et al., 2016*; *Kline et al., 2002*; *Peng et al., 2011*). Nevertheless, in both settings *Epas1* (not *Hif1a*) inactivation, had major effects on the proliferative response of the CBs to protracted hypoxia (*Hodson et al., 2016*).

Transcriptome analysis has confirmed that *Epas1* is the second most abundant transcript in neonatal CB glomus cells (*Tian et al., 1998*; *Zhou et al., 2016*), and that it is expressed at far higher levels than are found in cells of similar developmental origins, including superior cervical ganglion (SCG) sympathetic neurons (*Gao et al., 2017*). It has also been recently shown that *Epas1* but not *Hif1a*, overexpression in sympathoadrenal cells leads to enlargement of the CB (*Macías et al., 2014*). Here, we report that *Epas1* is required for the development of CB $O_2$-sensitive glomus cells, and that mutant animals lacking CB function have impaired adaptive physiological responses.

## Results

### Sympathoadrenal *Epas1* loss blocks carotid body glomus cell development

To elucidate the role of HIFα isoforms in CB development and function, we generated mouse strains carrying *Hif1a* or *Epas1* embryonic deletions (TH-HIF-1α$^{KO}$ and TH-HIF-2α$^{KO}$) restricted to catecholaminergic tissues by crossing them with a mouse strain expressing cre recombinase under the control of the endogenous tyrosine hydroxylase (*Th*) promoter (*Macías et al., 2014*).

Histological analysis of the carotid artery bifurcation dissected from adult (8–12 weeks old) TH-HIF-2α$^{KO}$ mice revealed virtually no CB TH$^+$ glomus cells (*Figure 1A*, bottom panels) compared to HIF-2α$^{WT}$ littermate controls (*Figure 1A*, top panels). Conversely, TH-HIF-1α$^{KO}$ mice showed a typical glomus cell organization, characterized by scattered clusters of TH$^+$ cells throughout the CB parenchyma (*Figure 1B*, bottom panels) similar to those found in HIF-1α$^{WT}$ littermate controls (*Figure 1B*, top panels). These observations were further corroborated by quantification of the total CB parenchyma volume (*Figure 1D–1F*).

Other catecholaminergic organs whose embryological origins are similar to those of the CB, for example, the superior cervical ganglion (SCG), do not show significant differences in structure or volume between TH-HIF-2α$^{KO}$ mutants and control mice (*Figure 1A–C*). This suggests a specific role of HIF-2α in the development of the CB glomus cells, and argues against a global role for the gene in late development of catecholaminergic tissues. Further evidence for this comes from phenotypic characterization of adrenal medulla (AM) TH$^+$ chromaffin cells. No major histological alterations were observed in adrenal glands removed from TH-HIF-2α$^{KO}$ and TH-HIF-1α$^{KO}$ mutant mice compared to HIF-2α$^{WT}$ and HIF-1α$^{WT}$ littermate controls (*Figure 1G and H*). Consistent with this, the amount of catecholamine (adrenaline and noradrenaline) present in the urine of TH-HIF-2α$^{KO}$ and TH-HIF-1α$^{KO}$ deficient mice was similar to that found in their respective littermate controls (*Figure 1I*).

To determine deletion frequencies in these tissues, we crossed a loxP-flanked Td-Tomato reporter strain (*Madisen et al., 2010*) with *Th-IRES-Cre* mice. Td-Tomato$^+$ signal was only detected within the CB, SCG and AM of mice expressing cre recombinase under the control of the *Th* promoter (*Figure 1—figure supplement 1A*). Additionally, SCG and AM from HIF-2α$^{WT}$, HIF-1α$^{WT}$, TH-HIF-2α$^{KO}$ and TH-HIF-1α$^{KO}$ were quantified for *Epas1* and *Hif1a* deletion efficiency using genomic DNA. As expected, there is a significant level of deletion of *Epas1* and *Hif1a* genes detected in TH-HIF-2α$^{KO}$ and TH-HIF-1α$^{KO}$ mutant mice compared to HIF-2α$^{WT}$ and HIF-1α$^{WT}$ littermate controls (*Figure 1—figure supplement 1B and C*).

To determine whether absence of CB glomus cells in adult TH-HIF-2α$^{KO}$ mice is a result of impaired glomus cell differentiation or cell survival, we examined carotid artery bifurcations dissected from TH-HIF-2α$^{KO}$ mice at embryonic stage E18.5 (i.e., 1–2 days before birth), and postnatally

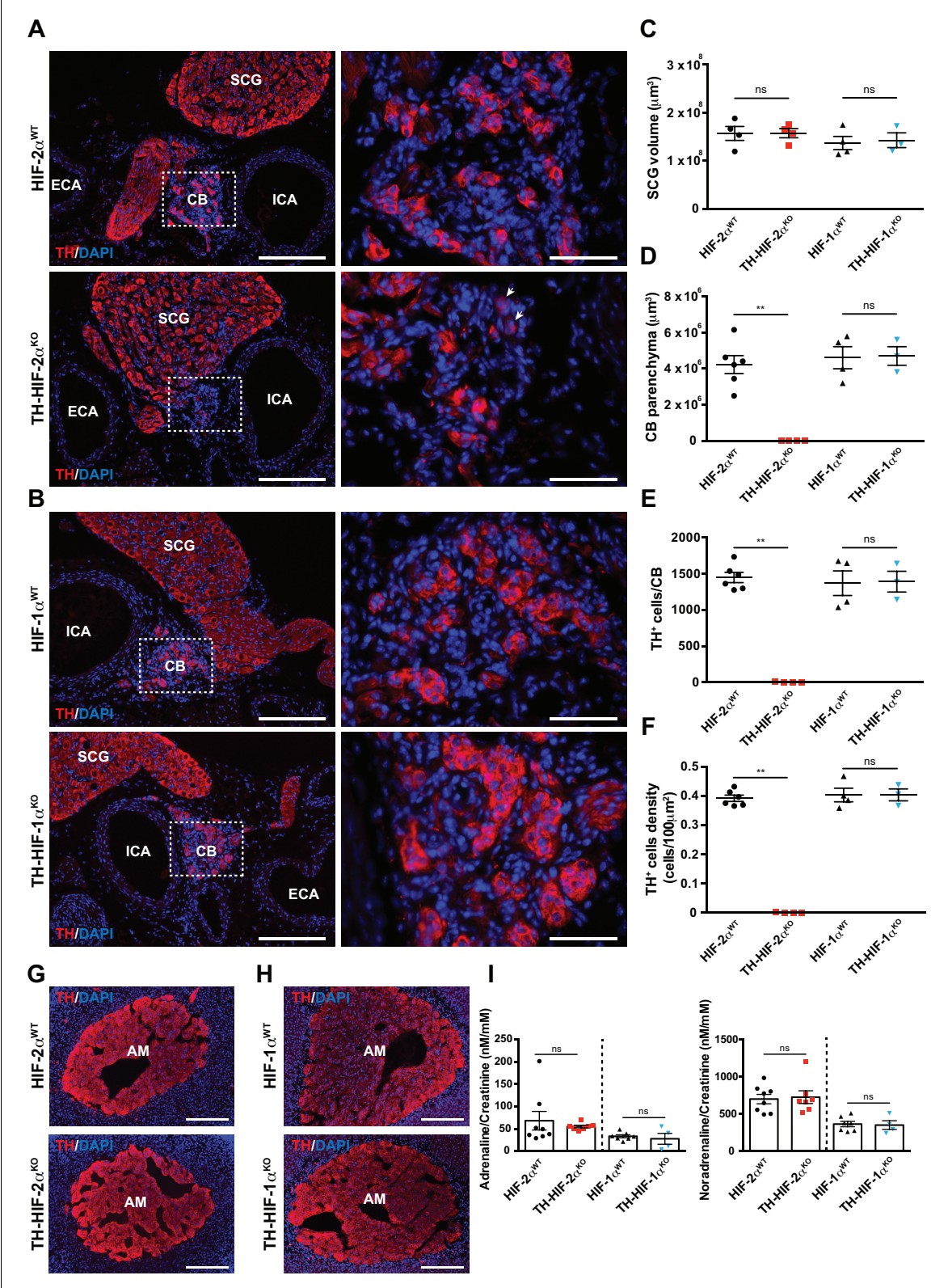

**Figure 1.** Selective loss of carotid body glomus cells in sympathoadrenal-specific *Epas1*, but not *Hif1a*, deficient mice. (**A and B**) Tyrosine hydroxylase (TH) immunostaining on carotid bifurcation sections from HIF-2α$^{WT}$ (A, top panels), TH-HIF-2α$^{KO}$ (A, bottom panels), HIF-1α$^{WT}$ (B, top panels) and TH-HIF-1α$^{KO}$ (B, bottom panels) mice (8–12 weeks old). Micrographs showed in A (bottom) were selected to illustrate the rare presence of TH$^+$ glomus cells in TH-HIF-2α$^{KO}$ mice (white arrowheads). Dashed rectangles (left panels) are shown at higher magnification on the right panels. SCG, superior cervical

*Figure 1 continued on next page*

*Figure 1 continued*

ganglion; CB, carotid body; ICA, internal carotid artery; ECA, external carotid artery. Scale bars: 200 µm (left panels), 50 µm (right panels). (**C–F**) Quantification of total SCG volume (**C**), total CB volume (**D**), TH[+] cell number (**E**) and TH[+] cell density (**F**) on micrograph from HIF-2α^WT(black dots, n = 4 for SCG and n = 6 for CB), TH-HIF-2α^KO (red squares, n = 4), HIF-1α^WT (black triangles, n = 4) and TH-HIF-1α^KO (blue triangles, n = 3) mice. Data are expressed as mean ± SEM. Mann-Whitney test, **p<0.001; ns, non-significant. (**G and H**) TH-immunostained adrenal gland sections from HIF-2α^WT(G, top panel), TH-HIF-2α^KO (G, bottom panel), HIF-1α^WT (H, top panel) and TH-HIF-1α^KO (H, bottom panel) littermates. AM, adrenal medulla. Scale bars: 200 µm. (**I**) Normalized adrenaline (left) and noradrenaline (right) urine content measured by ELISA. HIF-2α^WT(black dots, n = 8), TH-HIF-2α^KO (red squares, n = 7), HIF-1α^WT (black triangles, n = 7) and TH-HIF-1α^KO (blue triangles, n = 4). Mann-Whitney test, ns, non-significant.

DOI: https://doi.org/10.7554/eLife.34681.002

The following figure supplement is available for figure 1:

**Figure supplement 1.** Sympathoadrenal gene deletion by *Th-IRES-Cre* mouse strain.

DOI: https://doi.org/10.7554/eLife.34681.003

(P0) (*Figure 2A*). Differentiated TH[+] glomus cells were found in the carotid bifurcation of TH-HIF-2α^KO mutant mice at both stages (*Figure 2A*). However, there is a significant reduction in the total CB parenchyma volume and in the number of differentiated TH[+] glomus cells in TH-HIF-2α^KO mice at E18.5, and this reduction is even more evident in newborn mice (*Figure 2B and C*). Since CB volume and cells number of mutant mice decreased in parallel, no significant changes were seen in TH[+] cell density at these two time points, however (*Figure 2D*).

Consistent with the results described above (*Figure 1C*), the SCG volume was not different in mutant newborns relative to wild type controls (*Figure 2E*). We next assessed cell death in the developing CB, and found a progressive increase in the number of TUNEL[+] cells within the CB parenchyma of TH-HIF-2α^KO mice compared to HIF-2α^WT littermates (*Figure 2F and G*). These data demonstrate that HIF-2α, but not HIF-1α, is essential for the survival of CB glomus cells.

## Impaired ventilatory response and whole-body metabolic activity in TH-HIF-2α^KO mice exposed to hypoxia

To study the impact of the TH-HIF-2α^KO mutation on respiratory function, we examined ventilation in normoxia (21% O₂) and in response to environmental normobaric hypoxia (10% O₂) via whole-body plethysmography. *Figure 3A* illustrates the changes of respiratory rate during a prolonged hypoxic time-course in HIF-2α^WT and TH-HIF-2α^KO littermates. HIF-2α^WT mice breathing 10% O₂ had a biphasic response, characterized by an initial rapid increase in respiratory rate followed by a slow decline. In contrast, TH-HIF-2α^KO mice, which have normal ventilation rates in normoxia (*Figure 3A–C*), respond abnormally throughout the hypoxic period. Averaged respiratory rate and minute volume during the initial 5 min of hypoxia show a substantial reduction in ventilation rate in TH-HIF-2α^KO mutant mice relative to HIF-2α^WT controls (*Figure 3B and C*). However, the respiratory response to hypercapnia (5% CO₂) was preserved in TH-HIF-2α^KO mice, indicating normal function of the respiratory center (*Figure 3B and C*).

Additional respiratory parameters altered by the lack of CB in response to hypoxia are shown in *Figure 3—figure supplement 1*. Consistent with a failed HVR, hemoglobin saturation levels dropped to 50–55% in TH-HIF-2α^KO mice, as compared to 73% saturation in control animals after exposure to 10% O₂ for 5 min (*Figure 3D*). However, HIF-2α^WT and TH-HIF-2α^KO mice showed fully saturated hemoglobin levels while breathing 21% O₂ (*Figure 3H*). HIF-1α^WT and TH-HIF-1α^KO mice have similar respiratory responses to 21% O₂ and 10% O₂, which indicates that HIF-1α is not required for the CB-mediated ventilatory response to hypoxia (*Figure 3E–G*; *Figure 3—figure supplement 1F–J*).

We next determined metabolic activity in HIF-2α^WT and TH-HIF-2α^KO littermates in a 21% O₂ or 10% O₂ environment. Circadian metabolic profiles (VO₂, VCO₂ and RER) were similar in WT and TH-HIF-2α^KO mutant mice (*Figure 3I–K*). After exposure to hypoxia, however, TH-HIF-2α^KO mice showed lower VO₂ and VCO₂ peaks and had a significantly hampered metabolic adaptation to low oxygen (*Figure 3L and M*). There is no shift in substrate usage, as the respiratory exchange ratio (RER) was similar in HIF-2α^WT and TH-HIF-2α^KO littermates exposed to 10% O₂ (*Figure 3N*).

Although TH-HIF-2α^KO deficient mice did not show an apparent phenotype in the AM or SCG, it was important to determine whether their secretory or electrical properties were altered (*Figure 3—figure supplement 2*). The secretory activity of chromaffin cells under basal conditions and in

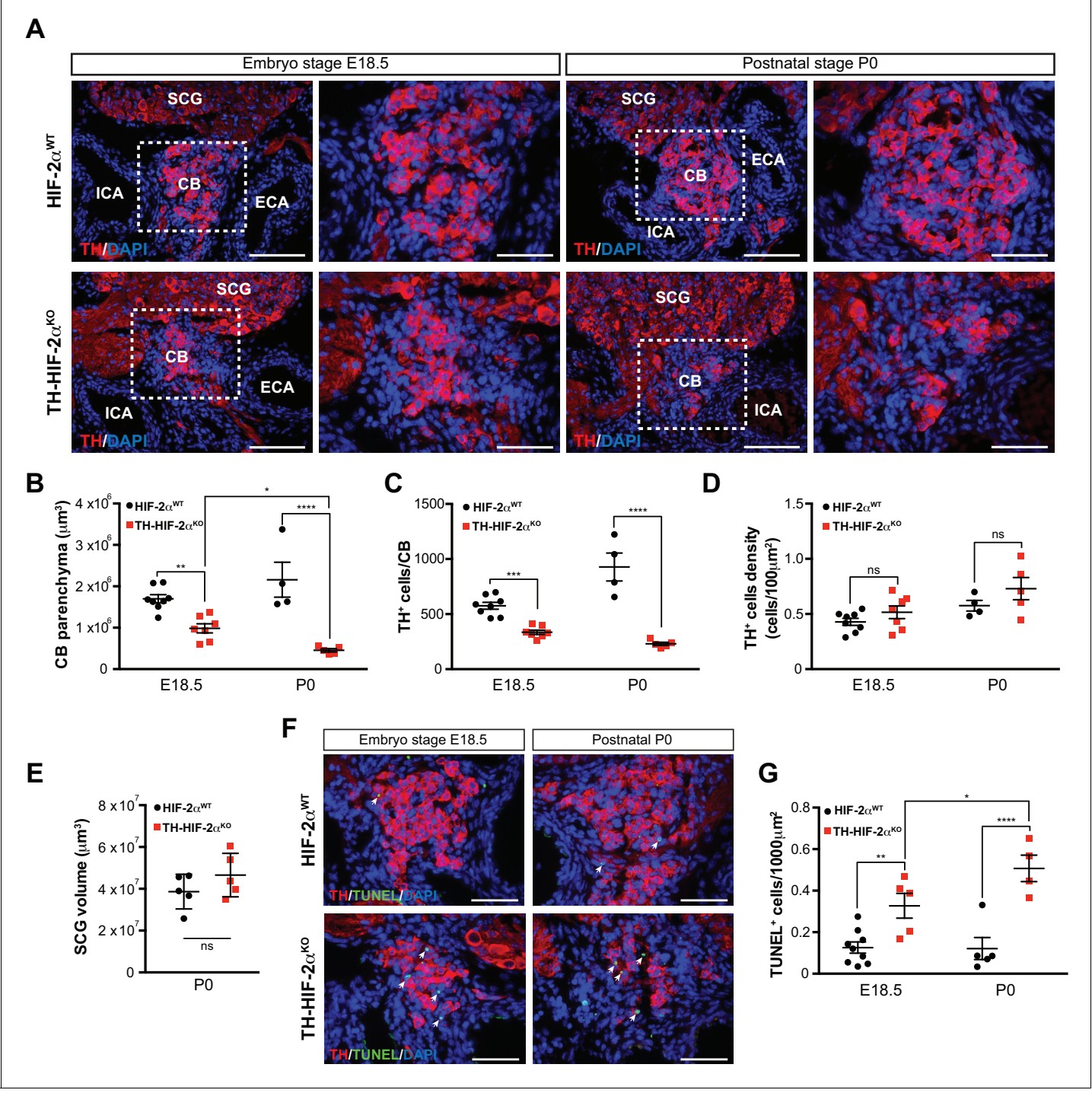

**Figure 2.** Progressive CB glomus cells death during development of TH-HIF-2α$^{KO}$ mouse. (**A**) Representative micrographs illustrating CB TH$^+$ glomus cells appearance at embryo stage E18.5 (left) and postnatal stage P0 (right) in HIF-2α$^{WT}$(top panels) and TH-HIF-2α$^{KO}$ (bottom panels) mice. Dashed rectangles (left panels) are shown at higher magnification on the right panels. SCG, superior cervical ganglion; CB, carotid body; ICA, internal carotid artery; ECA, external carotid artery. Scale bars: 100 µm. (**B–E**) Quantitative histological analysis of CB volume (**B**), TH$^+$ cells number (**C**), TH$^+$ cell density (**D**) and SCG volume (**E**) from TH-HIF-2α$^{KO}$ compared to HIF-2α$^{WT}$mice. HIF-2α$^{WT}$(E18.5, n = 8; P0, n = 4; SCG, n = 5), TH-HIF-2α$^{KO}$ (E18.5, n = 7; P0, n = 5; SCG, n = 5). SCG, superior cervical ganglion. Data are expressed as mean ±SEM. Two-way ANOVA, *p<0.05, **p<0.01, ***p<0.001, ****p<0.0001, ns, non significant. (**F**) Representative pictures of double TH$^+$ and TUNEL$^+$ (white arrows) stained carotid bifurcation sections from HIF-2α$^{WT}$(top panels) and TH-HIF-2α$^{KO}$ (bottom panels) mice at E18.5 (left) and P0 (right) stages. (**G**) Number of TUNEL$^+$ cells within the CB (TH$^+$ area) of HIF-2α$^{WT}$(E18.5, n = 9; P0, n = 5) and TH-HIF-2α$^{KO}$ (E18.5, n = 5; P0, n = 4) mice at E18.5 and P0 stages. Data are expressed as mean ±SEM. Two-way ANOVA, *p<0.05, **p<0.01, ****p<0.0001. Scale bars: 50 µm.

*Figure 2 continued on next page*

Figure 2 continued

DOI: https://doi.org/10.7554/eLife.34681.004

response to hypoxia, hypercapnia and high (20 mM) extracellular $K^+$ was not changed in TH-HIF-$2\alpha^{KO}$ mice (8–12 weeks old) relative to control littermates (*Figure 3—figure supplement 2A–D*). Current-voltage (I-V) relationships for $Ca^{2+}$ and $K^+$ currents of dispersed chromaffin cells (*Figure 3—figure supplement 2E*) and SCG neurons (*Figure 3—figure supplement 2F*) were also similar in HIF-$2\alpha^{WT}$ and TH-HIF-$2\alpha^{KO}$ animals. These data further support the notion that HIF-$2\alpha$ loss in the sympathoadrenal lineages predominantly affects CB glomus cells.

## Deficient acclimatization to chronic hypoxia in mice with CBs loss

The CB grows and expands during sustained hypoxia, due to the proliferative response of glia-like (GFAP$^+$) neural crest-derived stem cells within the CB parenchyma (*Arias-Stella and Valcarcel, 1976*; *Pardal et al., 2007*). Despite the absence of adult TH$^+$ oxygen sensing cells in TH-HIF-$2\alpha^{KO}$ mice, we identified a normal number of CB GFAP$^+$ stem cells in the carotid artery bifurcation of these animals (*Figure 4—figure supplement 1A*). We then determined the rate at which CB progenitors from TH-HIF-$2\alpha^{KO}$ mice were able to give rise to newly formed CB TH$^+$ glomus induced by chronic hypoxia in vivo. Typical CB enlargement and emergence of double BrdU$^+$ TH$^+$ glomus cells was seen in HIF-$2\alpha^{WT}$ mice after 14 days breathing 10% $O_2$ (*Figure 4—figure supplement 1B–D*), as previously described (*Pardal et al., 2007*). However, no newly formed double BrdU$^+$ TH$^+$ glomus cells were observed in TH-HIF-$2\alpha^{KO}$ mice after 14 days in hypoxia (*Figure 4—figure supplement 1B–D*). These observations are consistent with the documented role of CB glomus cells as activators of the CB progenitors proliferation in response to hypoxia (*Platero-Luengo et al., 2014*).

HIF-$2\alpha^{WT}$, HIF-$1\alpha^{WT}$ and TH-HIF-$1\alpha^{KO}$ mice adapted well and survived up to 21 days in a 10% $O_2$ atmosphere normally. However, long-term survival of the TH-HIF-$2\alpha^{KO}$ deficient mice kept in a 10% $O_2$ environment was severely compromised, and only 40% reached the experimental endpoint (*Figure 4A*). Mouse necropsies revealed ascites, congestive hepatomegaly and internal haemorrhages in the gut, which are most likely due to right ventricular failure. TH-HIF-$2\alpha^{KO}$ mice also showed increased haematocrit and haemoglobin levels relative to HIF-$2\alpha^{WT}$littermates when maintained for 21 days in a 10% $O_2$ atmosphere (*Figure 4B and C*). Additionally, platelet counts were significantly reduced, whereas lymphocytes, granulocytes and monocytes were unchanged in TH-HIF-$2\alpha^{KO}$ mice compared to HIF-$2\alpha^{WT}$littermates (*Figure 4—figure supplement 1E–H*). Consistent with this, after 21 days in hypoxia, heart and spleen removed from TH-HIF-$2\alpha^{KO}$ mice were enlarged relative to HIF-$2\alpha^{WT}$littermates, likely due to increased blood viscosity and extramedullary hematopoiesis, respectively (*Figure 4D–G*). TH-HIF-$1\alpha^{KO}$ mice did not show significant differences compared to HIF-$1\alpha^{WT}$ littermates maintained either in a 21% $O_2$ or 10% $O_2$ environment (*Figure 4—figure supplement 2A–D*).

Chronic exposure to hypoxia induces pulmonary arterial hypertension (PAH) in mice (*Cowburn et al., 2016*; *Stenmark et al., 2009*). The right ventricle was found to be significantly enlarged in TH-HIF-$2\alpha^{KO}$ mice relative to control mice after 21 days of 10% $O_2$ exposure, indicating an aggravation of pulmonary hypertension (*Figure 4H*). This result was further confirmed by direct measurement of right ventricular systolic pressure (RVSP) on anaesthetised mice (*Figure 4I*). Lung histology of mice chronically exposed to 10% $O_2$ (*Figure 4J*) shows that the percentage of fully muscularized lung peripheral vessels is significantly increased in TH-HIF-$2\alpha^{KO}$ mice relative to controls. The tunica media of lung peripheral vessels was also significantly thicker in TH-HIF-$2\alpha^{KO}$ mice compared to HIF-$2\alpha^{WT}$ littermates after 21 days of hypoxic exposure (*Figure 4K*). Taken together, these data demonstrate that adaptation to chronic hypoxia is highly reliant on a normally functioning CB.

## Cardiovascular homeostasis is altered in mice lacking CBs

An important element of the CB chemoreflex is an increase in sympathetic outflow (*López-Barneo et al., 2016b*). To determine whether this had been affected in TH-HIF-$2\alpha^{KO}$ mutants, cardiovascular parameters were recorded on unrestrained mice by radiotelemetry. Circadian blood pressure monitoring showed a significantly hypotensive condition that was more pronounced during the diurnal period (*Figure 5A and B*) and did not correlate with changes in heart rate, subcutaneous

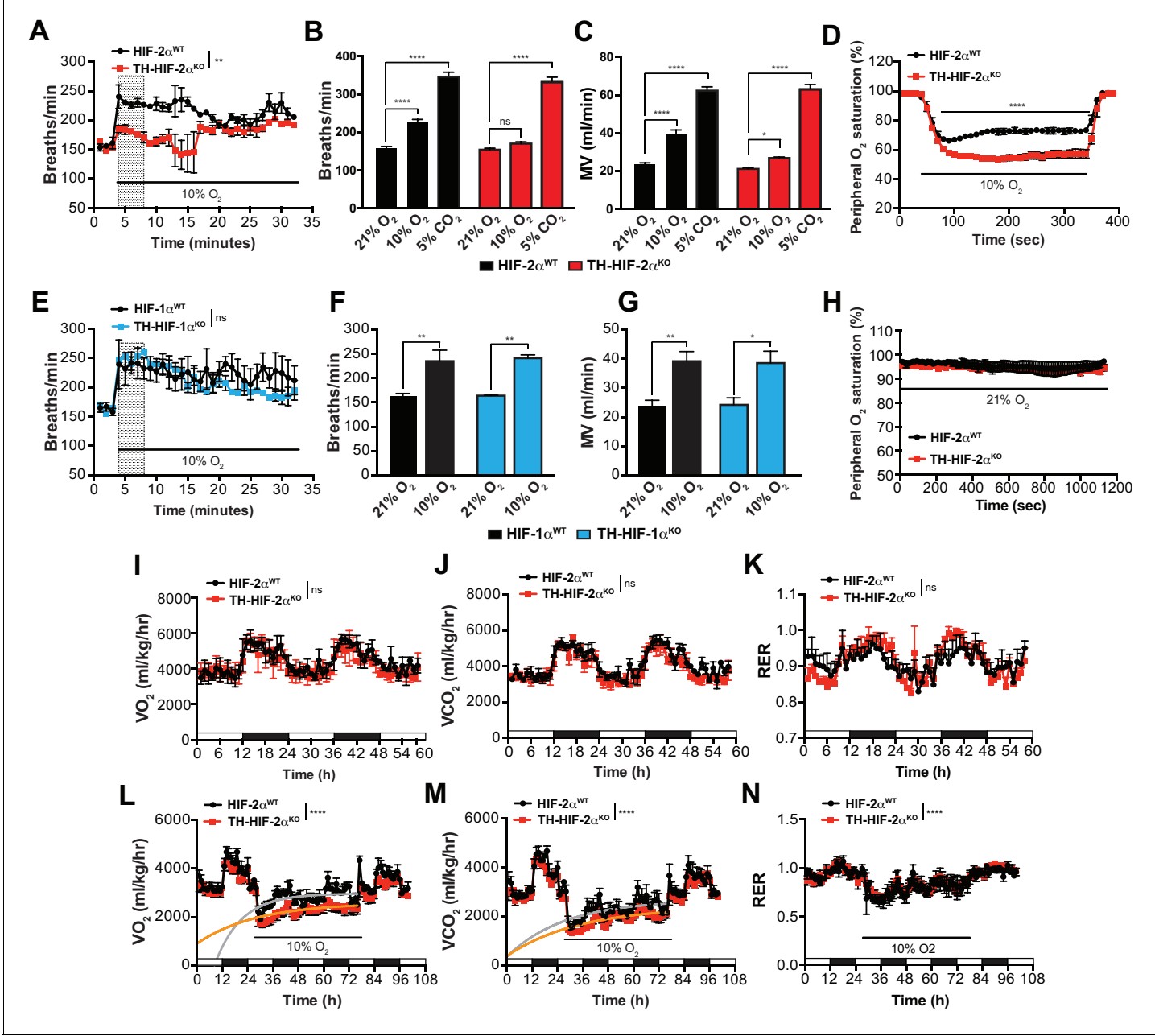

**Figure 3.** Effects of HIF-2α and HIF-1α sympathoadrenal loss on the HVR and whole-body metabolic activity in response to hypoxia. (**A and E**) HVR in TH-HIF-2α$^{KO}$ (A, red line, n = 4) and TH-HIF-1α$^{KO}$ (E, blue line, n = 3) deficient mice compared to control HIF-2α$^{WT}$(A, black line, n = 4) and HIF-1α$^{WT}$ (E, black line, n = 3) mice. Shift and duration of 10% O$_2$ stimulus (30 min) are indicated. Shaded areas represent the period of time analysed in B and C or F and G, respectively. Data are presented as mean breaths/min every minute ±SEM. Two-way ANOVA, **p<0.01, ns, non-significant. (**B and C**) Averaged respiratory rate (**B**) and minute volume (**C**) of HIF-2α$^{WT}$(black, n = 4) and TH-HIF-2α$^{KO}$ (red, n = 4) mice before and during the first 5 min of 10% O$_2$ (shaded rectangle in A) or 5% CO$_2$ exposure. Data are expressed as mean ±SEM. Two-way ANOVA, *p<0.05, ***p<0.001, ****p<0.0001, ns, non-significant. (**D and H**) Peripheral O$_2$ saturation before, during and after 10% O$_2$ (D) and 21% O$_2$ (H) in HIF-2α$^{WT}$(black, n = 8 in D, n = 4 in H) and TH-HIF-2α$^{KO}$ (red, n = 9 in D, n = 4 in H) mice. Data are expressed as mean percentage every 10 s ± SEM. Two-way ANOVA, ****p<0.0001. (**F and G**) Averaged respiratory rate (**F**) and minute volume (**F**) of HIF-1α$^{WT}$ (black, n = 3) and TH-HIF-1α$^{KO}$ (blue, n = 3) mice before and during the first 5 min of 10% O$_2$ exposure (shaded rectangle in E). Data are expressed as mean ±SEM. Two-way ANOVA, *p<0.05, **p<0.01. (**I–K**) Baseline metabolic activity of HIF-2α$^{WT}$(black, n = 3) and TH-HIF-2α$^{KO}$ (red, n = 3) mice. Data are presented as mean oxygen consumption (VO$_2$, (I), carbon dioxide generation (VCO$_2$, (J) and respiratory exchange ratio (RER, (K) every hour ±SEM. White and black boxes depict diurnal and nocturnal periods, respectively. Area under the curve followed by Mann-Whitney test, ns, non-significant. (**L–N**) Whole-body metabolic analysis of HIF-2α$^{WT}$(black, n = 4) and TH-HIF-2α$^{KO}$ (red, n = 6) mice in response to 10% O$_2$. Shift and duration of 10% O$_2$ stimulus are indicated. Data are presented as mean oxygen consumption (VO$_2$,

*Figure 3 continued on next page*

*Figure 3 continued*

(L), carbon dioxide generation (VCO$_2$, (M) and respiratory exchange ratio (RER, (N) every hour ±SEM. White and black boxes depict diurnal and nocturnal periods, respectively. Recovery across the hypoxia period was analysed by one-phase association curve fitting. ****p<0.0001.

DOI: https://doi.org/10.7554/eLife.34681.005

The following figure supplements are available for figure 3:

**Figure supplement 1.** Respiratory parameters of TH-HIF-2α$^{KO}$ and TH-HIF-1α$^{KO}$ mice exposed to acute hypoxia.

DOI: https://doi.org/10.7554/eLife.34681.006

**Figure supplement 2.** Secretory and electrophysiological properties in AM and SCG from TH-HIF-2α$^{KO}$ mice.

DOI: https://doi.org/10.7554/eLife.34681.007

temperature or activity (*Figure 5C*; *Figure 5—figure supplement 1A and B*). We next subjected radiotelemetry-implanted HIF-2α$^{WT}$ and TH-HIF-2α$^{KO}$ littermates to a 10% O$_2$ environment preceded and followed by a 24 hr period of normoxia (*Figure 5D–F*; *Figure 5—figure supplement 1C–E*). Normal control HIF-2α$^{WT}$ mice have a triphasic response to this level of hypoxia, characterised by a short (10 min) initial tachycardia and hypertension, followed by a marked drop both in heart rate and blood pressure; a partial to complete recovery is then seen after 24–36 hr (*Figure 5D and E*; *Figure 5—figure supplement 1C and D*)(*Cowburn et al., 2017*). However, while TH-HIF-2α$^{KO}$ mice experienced similar, but less deep, hypotension 3 hr post hypoxic challenge, they failed to recover, and showed a persistent hypotensive state throughout the hypoxic period (*Figure 5D*; *Figure 5—figure supplement 1C and D*). Interestingly, the bradycardic effect of hypoxia on heart rate was abolished in TH-HIF-2α$^{KO}$ animals (*Figure 5E*). Subcutaneous temperature, an indirect indicator of vascular resistance in the skin, showed an analogous trend to that seen in the blood pressure of TH-HIF-2α$^{KO}$ mutants (*Figure 5F*). Activity of TH-HIF-2α$^{KO}$ mice during hypoxia was slightly reduced relative to littermate controls (*Figure 5—figure supplement 1E*).

The carotid body has been proposed as a therapeutic target for neurogenic hypertension (*McBryde et al., 2013*; *Pijacka et al., 2016*). To determine the relationship between CB and hypertension in these mutants, we performed blood pressure recordings in an angiotensin II (Ang II)-induced experimental hypertension model (*Figure 5G*). As shown, TH-HIF-2α$^{KO}$ animals showed lower blood pressure relative to HIF-2α$^{WT}$ littermate controls after 7 days of Ang II infusion (*Figure 5H*; *Figure 5—figure supplement 1F and G*). Heart rate, subcutaneous temperature and physical activity were unchanged throughout the experiment (*Figure 5K–M*; *Figure 5—figure supplement 1*). Collectively, these data highlight the CB as a systemic cardiovascular regulator which, contributes to baseline blood pressure modulation, is critical for cardiovascular adaptations to hypoxia and delays the development of Ang II-induced experimental hypertension.

## Adaptive responses to exercise and high glucose are affected in mice with CB dysfunction

We next questioned whether this mutant, which lacks virtually all CB function, can inform questions about adaptation to other physiological stressors. In the first instance, we assessed metabolic activity during a maximal exertion exercise test. Oxygen consumption and carbon dioxide production (VO$_2$ and VCO$_2$, respectively) were comparable in HIF-2α$^{WT}$ and TH-HIF-2α$^{KO}$ mice at the beginning of the test. However, TH-HIF-2α$^{KO}$ mutants reached VO$_2$max much earlier than wild type controls, with significantly decreased heat generation (*Figure 6A,B and D*). There was no shift in energetic substrate usage, as evidenced by the lack of difference in respiratory exchange ratio (RER) (*Figure 6C*). There was a significant reduction in both aerobic capacity and power output relative to littermate controls (*Figure 6E–G*). Interestingly, although no significant changes were found in glucose blood content across the experimental groups and conditions (*Figure 6H*), lactate levels in TH-HIF-2α$^{KO}$ mice were significantly lower than those observed in HIF-2α$^{WT}$ mice after running (*Figure 6I*).

Carotid body function has been associated with glucose sensing (*Gao et al., 2014*; *Pardal and López-Barneo, 2002*). To determine whether this model was suitable for studying this phenomenon, we next examined the response of TH-HIF-2α$^{KO}$ mice to hyperglycemic and hypoglycemic challenge. Baseline glucose levels of HIF-2α$^{WT}$ and TH-HIF-2α$^{KO}$ animals were not statistically different under fed and fasting conditions (*Figure 6J and K*), and metabolic activity during fasting was unaffected (*Figure 6—figure supplement 1A–D*). However, TH-HIF-2α$^{KO}$ mutant mice showed significant

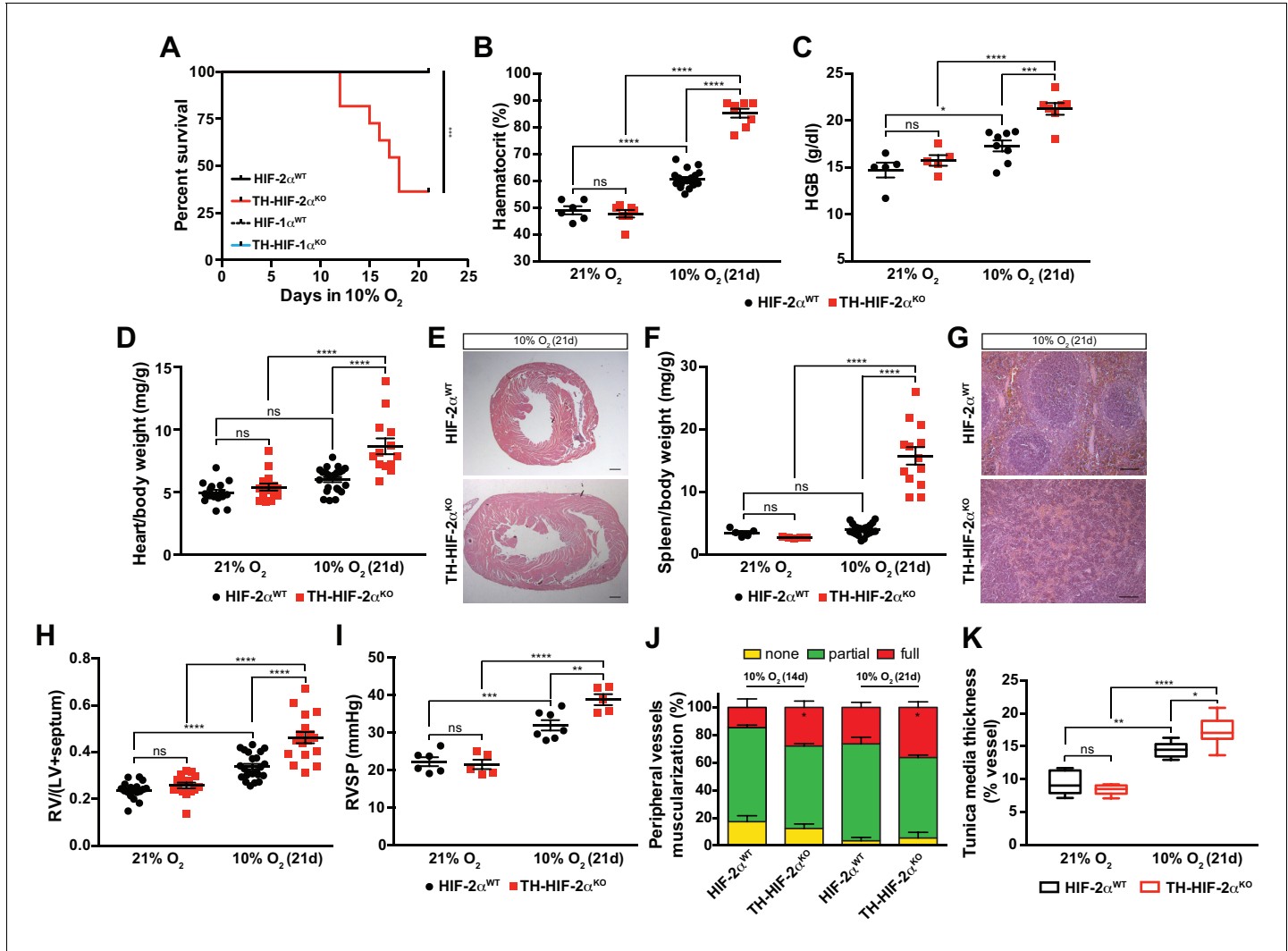

**Figure 4.** Impaired acclimatisation to chronic hypoxia and severe pulmonary hypertension in mice lacking CBs. (**A**) Kaplan-Meier survival curve of the indicated mice housed at 10% $O_2$ up to 21 days. HIF-2$\alpha^{WT}$(n = 12), TH-HIF-2$\alpha^{KO}$ (n = 11), HIF-1$\alpha^{WT}$ (n = 8) and TH-HIF-1$\alpha^{KO}$ (n = 8). ***p<0.001. (**B and C**) Haematocrit (**B**) and haemoglobin (**C**) levels of HIF-2$\alpha^{WT}$(black) and TH-HIF-2$\alpha^{KO}$ (red) littermates kept in normoxia (21% $O_2$) and after 21 days at 10% $O_2$. In B, HIF-2$\alpha^{WT}$(21% $O_2$, n = 6; 10% $O_2$, n = 18), TH-HIF-2$\alpha^{KO}$ (21% $O_2$, n = 7; 10% $O_2$, n = 8). In C, HIF-2$\alpha^{WT}$(21% $O_2$, n = 5; 10% $O_2$, n = 8), TH-HIF-2$\alpha^{KO}$ (21% $O_2$, n = 5; 10% $O_2$, n = 7). Data are expressed as mean ± SEM. Two-way ANOVA, *p<0.05, ***p<0.001, ****p<0.0001, ns, non-significant. (**D and F**) Normalized heart (**D**) and spleen (**F**) weight of HIF-2$\alpha^{WT}$(black) and TH-HIF-2$\alpha^{KO}$ (red) before and after 21 days in hypoxia (10% $O_2$). In D, HIF-2$\alpha^{WT}$(21% $O_2$, n = 16; 10% $O_2$, n = 23), TH-HIF-2$\alpha^{KO}$ (21% $O_2$, n = 15; 10% $O_2$, n = 13). In F, HIF-2$\alpha^{WT}$(21% $O_2$, n = 5; 10% $O_2$, n = 23), TH-HIF-2$\alpha^{KO}$ (21% $O_2$, n = 5; 10% $O_2$, n = 13). Data are expressed as mean ± SEM. Two-way ANOVA, ****p<0.0001, ns, non-significant. (**E and G**) Hematoxylin and eosin staining of heart (**E**) and spleen (**G**) of HIF-2$\alpha^{WT}$(top panel) and TH-HIF-2$\alpha^{KO}$ (bottom panel) after 21 days in hypoxia (10% $O_2$). Scale bars: 500 μm in E and 100 μm in G. (**H**) Fulton index on hearts dissected from HIF-2$\alpha^{WT}$(black, 21% $O_2$, n = 17; 10% $O_2$, n = 23) and TH-HIF-2$\alpha^{KO}$ (red, 21% $O_2$, n = 15; 10% $O_2$, n = 17) mice before and after 21 days in hypoxia (10% $O_2$). Data are expressed as mean ± SEM. Two-way ANOVA, ****p<0.0001, ns, non-significant. RV, right ventricle. LV, left ventricle. (**I**) Right ventricular systolic pressure (RVSP) recorded on HIF-2$\alpha^{WT}$(black, 21% $O_2$, n = 6; 10% $O_2$, n = 7) and TH-HIF-2$\alpha^{KO}$ (red, 21% $O_2$, n = 5; 10% $O_2$, n = 5) mice maintained in normoxia (21% $O_2$) or hypoxia (10% $O_2$) for 21 days. Data are expressed as mean ± SEM. Two-way ANOVA, **p<0.01, ***p<0.001, ****p<0.0001, ns, non-significant. (**J and K**) Lung peripheral vessels muscularization (**J**) and arterial medial thickness (**K**) in HIF-2$\alpha^{WT}$and TH-HIF-2$\alpha^{KO}$ mice exposed to 10% $O_2$ for 14 and/or 21 days. In H, HIF-2$\alpha^{WT}$(10% $O_2$ 14d, n = 4; 10% $O_2$ 21d, n = 3), TH-HIF-2$\alpha^{KO}$ (10% $O_2$ 14d, n = 6; 10% $O_2$ 21d, n = 3). In I, HIF-2$\alpha^{WT}$(21% $O_2$, n = 5; 10% $O_2$ 21d, n = 8), TH-HIF-2$\alpha^{KO}$ (21% $O_2$, n = 5; 10% $O_2$ 21d, n = 8). Data are expressed as mean ± SEM. Two-way ANOVA, *p<0.05, **p<0.01, ****p<0.0001, ns, non-significant.

DOI: https://doi.org/10.7554/eLife.34681.008

The following figure supplements are available for figure 4:

**Figure supplement 1.** Impaired hypoxia-induced CB proliferation in TH-HIF-2$\alpha^{KO}$ mice.
DOI: https://doi.org/10.7554/eLife.34681.009

**Figure supplement 2.** Adaptation to hypoxia in TH-HIF-1$\alpha^{KO}$ animals.

*Figure 4 continued on next page*

Figure 4 continued

DOI: https://doi.org/10.7554/eLife.34681.010

changes in glucose tolerance, with no change in insulin tolerance when compared to HIF-$2\alpha^{WT}$littermates (*Figure 6L and M*). TH-HIF-$2\alpha^{KO}$ mice also showed an initially hampered lactate clearance relative to control mice (*Figure 6N*). Together, these data demonstrate that the mutants demonstrate intriguing roles for the CB in metabolic adaptations triggered by both exercise and hyperglycemia.

## CB development and proliferation in response to hypoxia require mTORC1 activation

Hypoxia has differential effects on cell proliferation. While HIF-1$\alpha$ is known to promote cell arrest via mTORC1 repression and other mechanisms (*Brugarolas et al., 2004*; *Cam et al., 2010*; *Carmeliet et al., 1998*; *Goda et al., 2003*; *Koshiji et al., 2004*), HIF-2$\alpha$ can activate mTORC1 to induce cell proliferation and growth in a number of cell types and organs, for example, tumour cells or the lung epithelium (*Brusselmans et al., 2003*; *Cowburn et al., 2016*; *Elorza et al., 2012*; *Gordan et al., 2007*; *Hubbi and Semenza, 2015*; *Kondo et al., 2003*; *Raval et al., 2005*; *Torres-Capelli et al., 2016*). Physiological proliferation and differentiation of the CB induced by hypoxia closely resembles CB developmental expansion (*Annese et al., 2017*; *Pardal et al., 2007*; *Platero-Luengo et al., 2014*); therefore a potential explanation for the developmental defects in CB expansion observed in above could be defects in mTORC1 activation.

To test this hypothesis, we first determined mTORC1 activation levels in the CB of mice maintained at 21% $O_2$ or 10% $O_2$ for 14 days. This was done via immunofluorescent detection of the phosphorylated form of the ribosomal protein S6 (p-S6) (*Elorza et al., 2012*; *Ma and Blenis, 2009*). In normoxia, there was a faint p-S6 expression within the TH$^+$ glomus cells of the CB in wild type mice; this contrasts with the strong signal detected in SCG sympathetic neurons (*Figure 7A*, left panels; *Figure 7—figure supplement 1A*, left panel). However, after 14 days in a 10% $O_2$ environment, we saw a significantly increased p-S6 signal within the CB TH$^+$ cells while the SCG sympathetic neurons remained at levels similar to those seen in normoxia (*Figure 7A and B*; *Figure 7—figure supplement 1A and B*). TH expression, which is induced by hypoxia (*Schnell et al., 2003*), was also significantly elevated in CB TH$^+$ cells, and trended higher in SCG sympathetic neurons (*Figure 7B*; *Figure 7—figure supplement 1B*).

We next assessed the role of HIF-2$\alpha$ in this hypoxia-induced mTORC1 activation. We began with evaluating expression of p-S6 in late stage embryos. In E18.5 embryos there was a striking accumulation of p-S6; this is significantly reduced in HIF-2$\alpha$ deficient CB TH$^+$ cells in embryos from this stage of development (*Figure 7C and D*). p-S6 levels in SCG TH$^+$ sympathetic neurons of HIF-2$\alpha^{WT}$and TH-HIF-2$\alpha^{KO}$ E18.5 embryos were unaffected, however (*Figure 7—figure supplement 1C and D*). These data indicate that HIF-2$\alpha$ regulates mTORC1 activity in the CB glomus cells during development.

To determine whether this pathway could be pharmacologically manipulated, we then studied the effect of mTORC1 inhibition by rapamycin on CB proliferation and growth following exposure to environmental hypoxia. mTORC1 activity, as determined by p-S6 staining on SCG and CB TH$^+$ cells after 14 days of hypoxic treatment, was inhibited after in vivo rapamycin administration (*Figure 7—figure supplement 1E*). Interestingly, histological analysis of rapamycin injected wild type mice maintained for 14 days at 10% $O_2$ showed decreased CB parenchyma volume, TH$^+$ cells number, and showed a dramatic reduction in the number of proliferating double BrdU$^+$ TH$^+$ cells when compared to vehicle-injected mice (*Figure 7E–H*). Rapamycin treatment did not alter the phenotype previously observed in TH-HIF-2$\alpha^{KO}$ mice (*Figure 7E–H*; *Figure 4—figure supplement 1*). Additionally, rapamycin administration had no effect on CB parenchyma volume or TH$^+$ cell numbers of wild type mice breathing environmental oxygen (*Figure 7—figure supplement 1*).

In addition, we assessed the ventilatory acclimatization to hypoxia (VAH), as a progressive increase in ventilation rates is associated with CB growth during sustained exposure to hypoxia (*Hodson et al., 2016*). Consistent with the histological analysis above, rapamycin treated mice had a decreased VAH compared to vehicle-injected control mice following 7 days at 10% $O_2$ (*Figure 7I*).

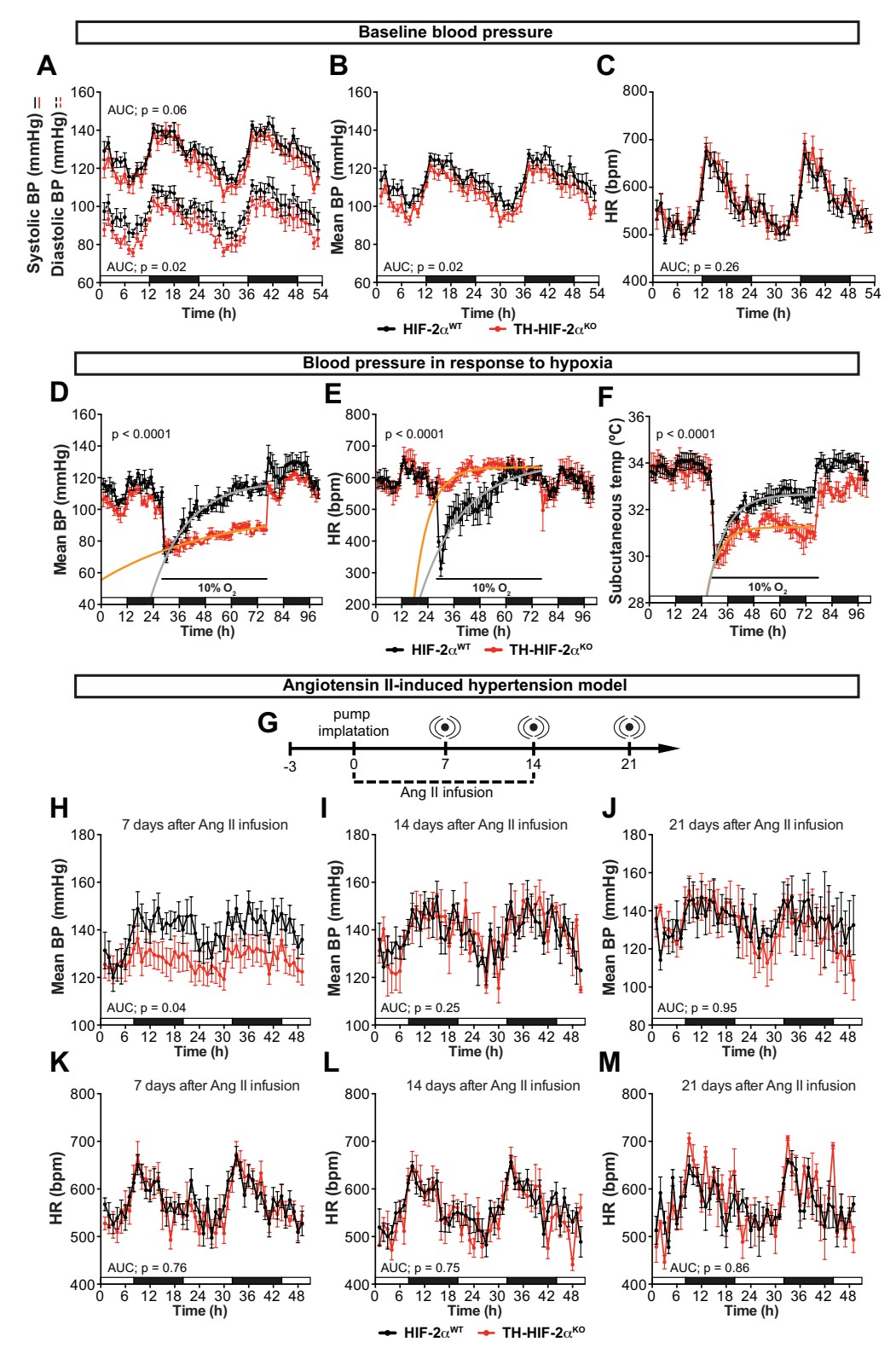

**Figure 5.** Effects of CB dysfunction on blood pressure regulation. (A–C) Circadian variations in baseline systolic blood pressure (A, continued line), diastolic blood pressure (A, broken line), mean blood pressure (B) and heart rate (C) of HIF-2α$^{WT}$(black, n = 10) and TH-HIF-2α$^{KO}$ (red, n = 8) littermates recorded by radiotelemetry. White and black boxes depict diurnal and nocturnal periods, respectively. Data are presented as averaged values every hour ±SEM. Area under the curve followed by unpaired t-test are shown. BP, blood pressure. HR, heart rate in beats per minute. (D–F) Mean blood

*Figure 5 continued on next page*

*Figure 5 continued*

pressure (D), heart rate (E) and subcutaneous temperature (F) alteration in response to hypoxia in radiotelemetry-implanted TH-HIF-2α$^{KO}$ (red, n = 5) compared to HIF-2α$^{WT}$(black, n = 5). Shift and duration of 10% $O_2$ are indicated in the graph. White and black boxes represent diurnal and nocturnal periods, respectively. Data are presented as averaged values every hour ±SEM. Recovery across the hypoxia period was analysed by one-phase association curve fitting. BP, blood pressure. HR, heart rate in beats per minute. (G) Schematic representation of the protocol followed in the Ang II-induced experimental hypertension model. (H–M) Mean blood pressure and heart rate recorded from HIF-2α$^{WT}$(black, n = 8) and TH-HIF-2α$^{KO}$ (red, n = 8) mice after 7 days (H and K), 14 days (I and L) and 21 days (J and M) of Ang II infusion. White and black boxes depict diurnal and nocturnal periods, respectively. Data are presented as averaged values every hour ±SEM. Area under the curve followed by unpaired t-test are shown. BP, blood pressure. HR, heart rate in beats per minute.

DOI: https://doi.org/10.7554/eLife.34681.011

The following figure supplement is available for figure 5:

**Figure supplement 1.** Cardiovascular homeostasis in response to hypoxia and Ang II-induced experimental hypertension.

DOI: https://doi.org/10.7554/eLife.34681.012

Failed ventilatory response to hypoxia was still observed in TH-HIF-2α$^{KO}$ mice after rapamycin treatment (*Figure 7I*). Haematocrit levels were comparable between vehicle control and rapamycing treated wild type mice after 7 days in hypoxia (*Figure 7—figure supplement 1F*).

Collectively, these data suggest that mTORC1 activation is required for CB glomus cell development, in a HIF-2α-dependent manner, and that this activation regulates CB growth as well as VAH during prolonged hypoxia.

## Discussion

Metabolic homeostasis depends on a complex system of $O_2$-sensing cells and organs. The carotid body is central to this, via its regulation of metabolic and ventilatory responses to oxygen (*López-Barneo et al., 2016a*). At the cellular level, the HIF transcription factor is equally central for responses to oxygen flux. However, how HIF contributes to the development and function of CB $O_2$-sensitivity is still not completely understood. Initial studies performed on global *Hif1a*$^{-/-}$ or *Epas1*$^{-/-}$ mutant mice revealed that HIFs are critical for development (*Compernolle et al., 2002*; *Iyer et al., 1998*; *Peng et al., 2000*; *Ryan et al., 1998*; *Scortegagna et al., 2003*; *Tian et al., 1998*), but only *Epas1*$^{-/-}$ embryos showed defects in catecholamine homeostasis, which indicated a potential role in sympathoadrenal development (*Tian et al., 1998*).

Investigations using hemizygous *Hif1a*$^{+/-}$ and *Epas1*$^{+/-}$ showed no changes in the CB histology or overall morphology (*Kline et al., 2002*; *Peng et al., 2011*). However, our findings demonstrate that embryological deletion of *Epas1*, but not *Hif1a*, specifically in sympathoadrenal tissues (TH$^+$) prevents $O_2$-sensitive glomus cell development, without affecting SCG sympathetic neurons or AM chromaffin cells of the same embryological origin. A potential explanation for this restricted effect of HIF-2α ablation in CB glomus cells development, as opposed to a more global effect on sympathoadrenal development (*Tian et al., 1998*) could be due to the developmental timing of Cre recombinase expression. However, at embryonic stage E13, when the CB is yet not developed, there are abundant numbers of TH-expressing cells both in the superior cervical ganglion (SCG) and the adrenal medulla (*Huber et al., 2002*; *Kameda, 2005*). Therefore, these results highlight the specific role of HIF-2α in CB glomus cell development, rather than a generalized effect on late development of the sympathoadrenal system.

It was previously reported that HIF-2α, but not HIF-1α overactivation in catecholaminergic tissues led to marked enlargement of the CB (*Macías et al., 2014*). Mouse models of Chuvash polycythemia, which have a global increase in HIF-2α levels, also show CB hypertrophy and enhanced acute hypoxic ventilatory response (*Slingo et al., 2014*). Thus either over-expression or deficiency of HIF-2α clearly perturbs CB development and, by extension, function.

Sustained hypoxia is generally thought to attenuate cell proliferation in a HIF-1α-dependent fashion (*Carmeliet et al., 1998*; *Goda et al., 2003*; *Koshiji et al., 2004*). However, environmental hypoxia can also trigger growth of a number of specialized tissues, including the CB (*Arias-Stella and Valcarcel, 1976*; *Pardal et al., 2007*) and the pulmonary arteries (*Madden et al., 1992*). In this context, the HIF-2α isoform seems to play the more prominent role, and has been associated with pulmonary vascular remodeling as well as with bronchial epithelial proliferation (*Brusselmans et al.,*

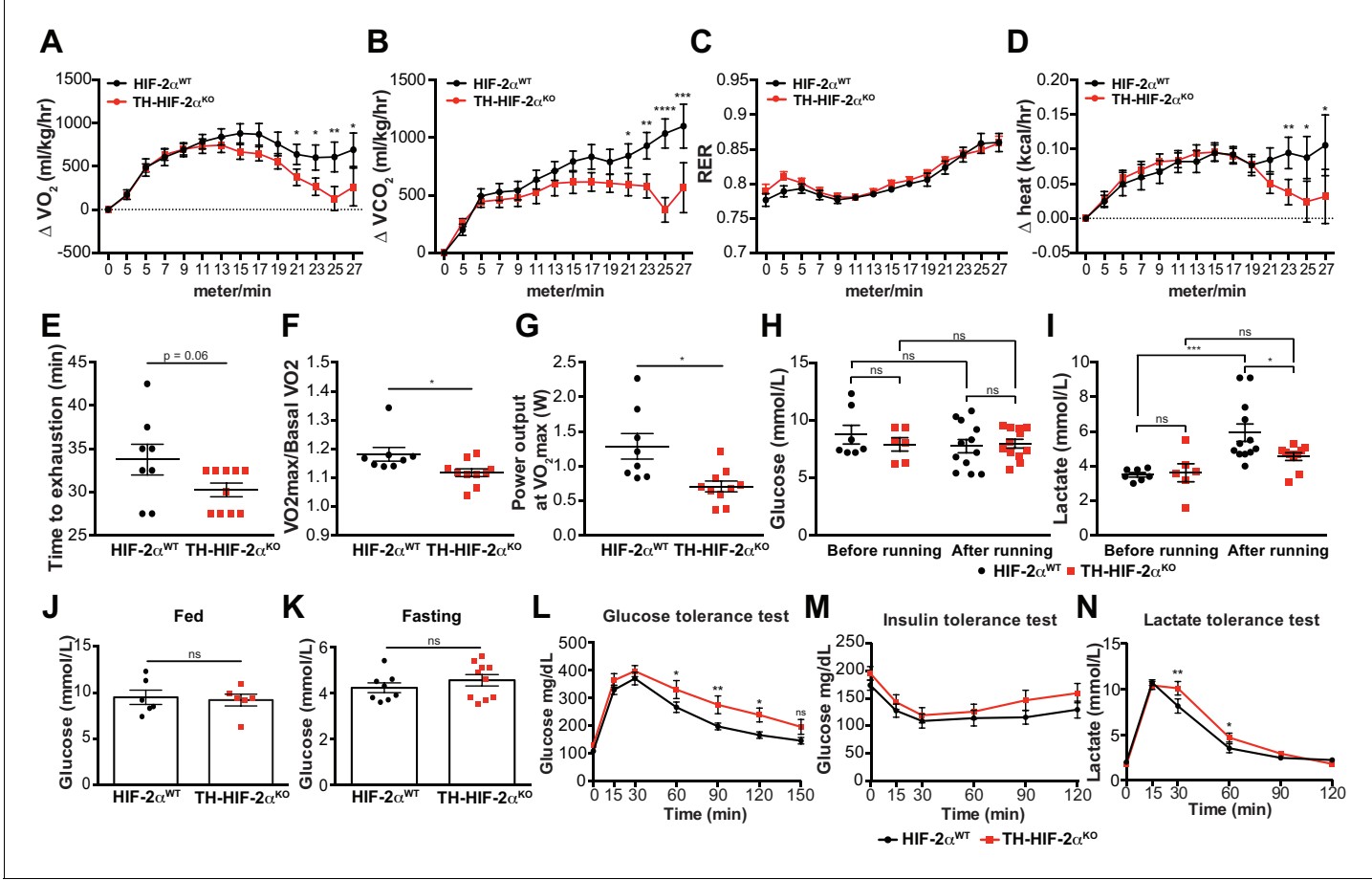

**Figure 6.** Exercise performance and glucose management in mice lacking CBs. (**A–D**). Time course showing the increment in $O_2$ consumption (**A**), increment in $CO_2$ generation (**B**), respiratory exchange ratio (**C**) and increment in heat production (**D**) from HIF-2α[WT](black, n = 8) and TH-HIF-2α[KO] (red, n = 10) mice during exertion on a treadmill. Mice were warmed up for 5 min (5 m/min) and then the speed was accelerated 2 m/min at 2.5 min intervals until exhaustion. Averaged measurements every 2.5 min intervals ± SEM are charted. Two-way ANOVA, *p<0.05, **p<0.01, ***p<0.001, ****p<0.0001. RER, respiratory exchange ratio. (**E**) Running total time spent by HIF-2α[WT](black, n = 8) and TH-HIF-2α[KO] (red, n = 10) mice to become exhausted. Data are expressed as mean ± SEM. Unpaired t-test. (**F and G**) Aerobic capacity (**F**) and power output (**G**) measured in HIF-2α[WT](black, n = 8) mice compared to TH-HIF-2α[KO] (red, n = 10) mice when performing at VO2max. Data are expressed as mean ± SEM. Unpaired t-test. *p<0.05. (**H and I**) Plasma glucose (**H**) and lactate (**I**) levels of HIF-2α[WT](black, before, n = 7; after, n = 12) and TH-HIF-2α[KO] (black, before, n = 6; after, n = 12) littermates before and after running to exhaustion. Data are expressed as mean ± SEM. Two-way ANOVA, *p<0.05. ***p<0.001. ns, non-significant. (**J and K**) Plasma glucose levels of HIF-2α[WT](black, n = 6) and TH-HIF-2α[KO] (red, n = 6) mice fed (**J**, n = 6 per genotype) or fasted over night (**K**, HIF-2α[WT](n = 8) and TH-HIF-2α[KO] (n = 10). Data are expressed as mean ± SEM. Unpaired t-test. (**L–M**) Time courses showing the tolerance of HIF-2α[WT]and TH-HIF-2α[KO] mice to glucose (**L**, n = 7 per genotype) insulin (**M**, n = 11 per genotype) and lactate (**N**, n = 7 per genotype) injected bolus. Data are presented as mean ± SEM. Two-way ANOVA, *p<0.05, **p<0.01, ns, non-significant.

DOI: https://doi.org/10.7554/eLife.34681.013

The following figure supplement is available for figure 6:

**Figure supplement 1.** Related to *Figure 6*.

DOI: https://doi.org/10.7554/eLife.34681.014

---

*2003*; *Bryant et al., 2016*; *Cowburn et al., 2016*; *Kapitsinou et al., 2016*; *Tan et al., 2013*; *Torres-Capelli et al., 2016*).

CB enlargement induced by hypoxia to a large extent recapitulates the cellular events that give rise to mature $O_2$-sensitive glomus cells in development. In a recent study, global HIF-2α, but not HIF-1α, inactivation in adult mice was shown to impair CB growth and associated VAH caused by exposure to chronic hypoxia (*Hodson et al., 2016*). Therefore, we used this physiological proliferative response as a model to understand the developmental mechanisms underpinning of HIF-2α deletion in $O_2$-sensitive glomus cells.

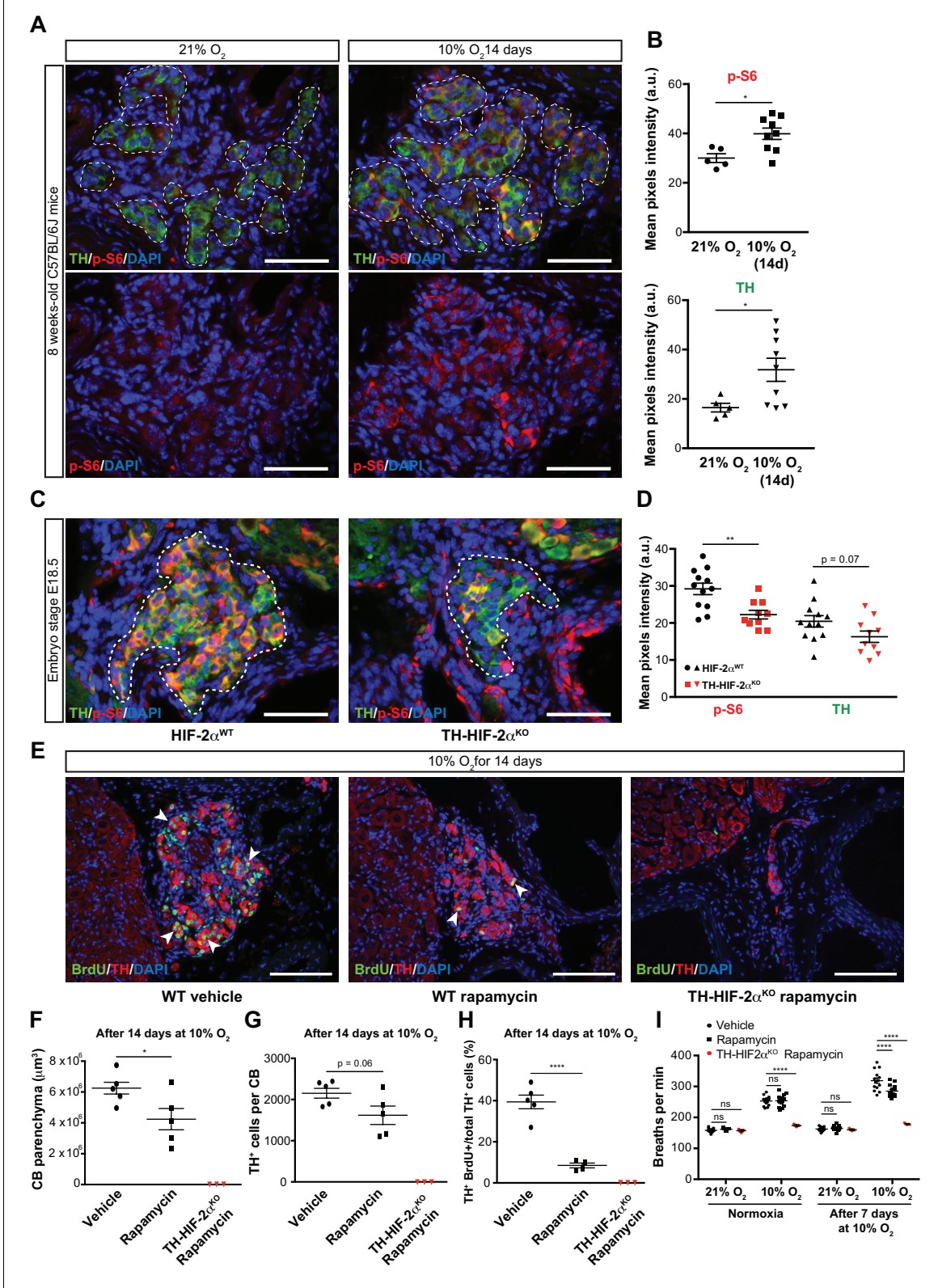

**Figure 7.** mTORC1 activation in CB development and hypoxia-induced CB neurogenesis. (**A**) Double TH and p-S6 (mTORC1 activation marker) immunofluorescence on adult wild type CB slices to illustrate the increased p-S6 expression after 14 days in hypoxia (10% O$_2$). For clarity, bottom panels only show p-S6 staining. TH$^+$ outlined area (top panels) indicates the region used for pixels quantification. Scale bars: 50 μm. (**B**) Quantification of p-S6 (top) and TH (bottom) mean pixels intensity within the CB TH$^+$ cells of wild type mice breathing 21% O$_2$ or 10% O$_2$ for 14 days. Each data point

*Figure 7 continued*

depicts the average of 3–4 pictures. 21% $O_2$, n = 5; 10% $O_2$ 14d, n = 9 per genotype. Data are presented as mean ± SEM. Unpaired t-test, *p<0.05. (C) Representative micrographs of TH and p-S6 double immunostaining of HIF-2α$^{WT}$ and TH-HIF-2α$^{KO}$ embryonic (E18.5) carotid bifurcations. TH$^+$ outlined area indicates the region used for pixels quantification. Scale bars: 50 µm. (D) Quantitative analysis performed on HIF-2α$^{WT}$(black, n = 12) and TH-HIF-2α$^{KO}$ (red, n = 10) micrographs previously immuno-stained for p-S6 and TH. Each data point depicts the average of 3–4 pictures. Data are presented as mean ± SEM. Unpaired t-test, **p<0.01. (E) Double BrdU and TH immunofluorescence on carotid bifurcation sections from wild type mice (left and central panels) or TH-HIF-2α$^{KO}$ mice (right panel) exposed to 10% $O_2$ and injected with rapamycin or vehicle control for 14 days. Notice the striking decrease in double BrdU$^+$ TH$^+$ cells (white arrowheads) in the rapamycin treated wild type mice compared to vehicle control animals. Scale bars: 100 µm. (F–H) Quantification of total CB volume (F), TH$^+$ cells number (G) and percentage of double BrdU$^+$ TH$^+$ over the total TH$^+$ cells (H) in wild type mice (black) or TH-HIF-2α$^{KO}$ mice (red) breathing 10% $O_2$ and injected with rapamycin or vehicle control for 14 days. Wild type, n = 5 mice per treatment. TH-HIF-2α$^{KO}$, n = 3. (I) Averaged breaths per minute during the first 2 min of acute hypoxia (10% $O_2$) exposure from vehicle or rapamycin injected wild type mice (black) or TH-HIF-2α$^{KO}$ (red) mice before and after 7 days of chronic hypoxia (10% $O_2$) treatment. Wild type, n = 14 mice per condition and treatment. TH-HIF-2α$^{KO}$, n = 3. Data are presented as mean ± SEM. B, D, F, G and H, unpaired t-test, *p<0.05, ****p<0.0001. I, two-way ANOVA followed by Tukey's multiple comparison test. ****p<0.0001, ns, non-significant.

DOI: https://doi.org/10.7554/eLife.34681.015

The following figure supplement is available for figure 7:

**Figure supplement 1.** mTORC1 activity in SCG TH$^+$ sympathetic neurons and effect of rapamycin treatment.
DOI: https://doi.org/10.7554/eLife.34681.016

Our findings reveal that mTORC1, a central regulator of cellular metabolism, proliferation and survival (*Laplante and Sabatini, 2012*), is necessary for CB $O_2$-sensitive glomus cell proliferation; and further, that impact in the subsequent enhancement of the hypoxic ventilatory response induced by chronic hypoxia. Importantly, we found that mTORC1 activity is reduced in HIF-2α-deficient embryonic glomus cells, but not in SCG sympathetic neurons, which likely explains the absence of a functional adult CB in these mutants.

Contrasting effects of HIF-α isoforms on the mammalian target of rapamycin complex 1 (mTORC1) activity can be found in the literature that mirror the differential regulation of cellular proliferation induced by hypoxia in different tissues (see above) (*Brugarolas et al., 2004*; *Brusselmans et al., 2003*; *Cam et al., 2010*; *Elorza et al., 2012*; *Liu et al., 2006*; *Torres-Capelli et al., 2016*). For instance, the activity of mTORC1 may be repressed via HIF-1α-induced gene expression of *Ddit4* or *Bnip3* (*Brugarolas et al., 2004*; *Li et al., 2007*). On the other hand, it has been shown that HIF-2α regulates mTORC1 activity in VHL-deficient tumour cells through the transcriptional control of the amino acid transporter *Slc7a5* (*Elorza et al., 2012*).

As HIF-2α regulates mTORC1 activity to promote proliferation, this connection could prove relevant in pathological situations; for example, both somatic and germline *EPAS1* gain-of-function mutations have been found in patients with paraganglioma, a catecholamine-secreting tumor of neural crest origin (*Lorenzo et al., 2013*; *Zhuang et al., 2012*). Indeed, CB tumors are 10 times more frequent in people living at high altitude (*Saldana et al., 1973*). As further evidence of this link, it has been shown that the mTOR pathway is commonly activated in both paraganglioma and pheochromocytoma tumors (*Du et al., 2016*; *Favier et al., 2015*).

CB chemoreception plays a crucial role in homeostasis, particularly in regulating cardiovascular and respiratory responses to hypoxia. Our data show that mice lacking CB $O_2$-sensitive cells (TH-HIF-2α$^{KO}$ mice) have impaired respiratory, cardiovascular and metabolic adaptive responses. Although previous studies have linked HIF-2α and CB function (*Hodson et al., 2016*; *Peng et al., 2011*), none of them exhibited a complete CB loss of function. This might explain the differing effects observed in hemizygous *Epas1*$^{+/-}$ mice (*Hodson et al., 2016*; *Peng et al., 2011*) compared to our TH-HIF-2α$^{KO}$ mice.

Mice with a specific *Hif1a* ablation in CB $O_2$-sensitive glomus cells (TH-HIF-1α$^{KO}$ mice) did not show altered hypoxic responses in our experimental conditions. Consistent with this, hemizygous *Hif1a*$^{+/-}$ mice exhibited a normal acute HVR (*Hodson et al., 2016*; *Kline et al., 2002*) albeit some otherwise aberrant CB activities in response to hypoxia (*Kline et al., 2002*; *Ortega-Sáenz et al., 2007*).

Alterations in CB development and/or function have recently acquired increasing potential clinical significance and have been associated with diseases such as heart failure, obstructive sleep apnea,

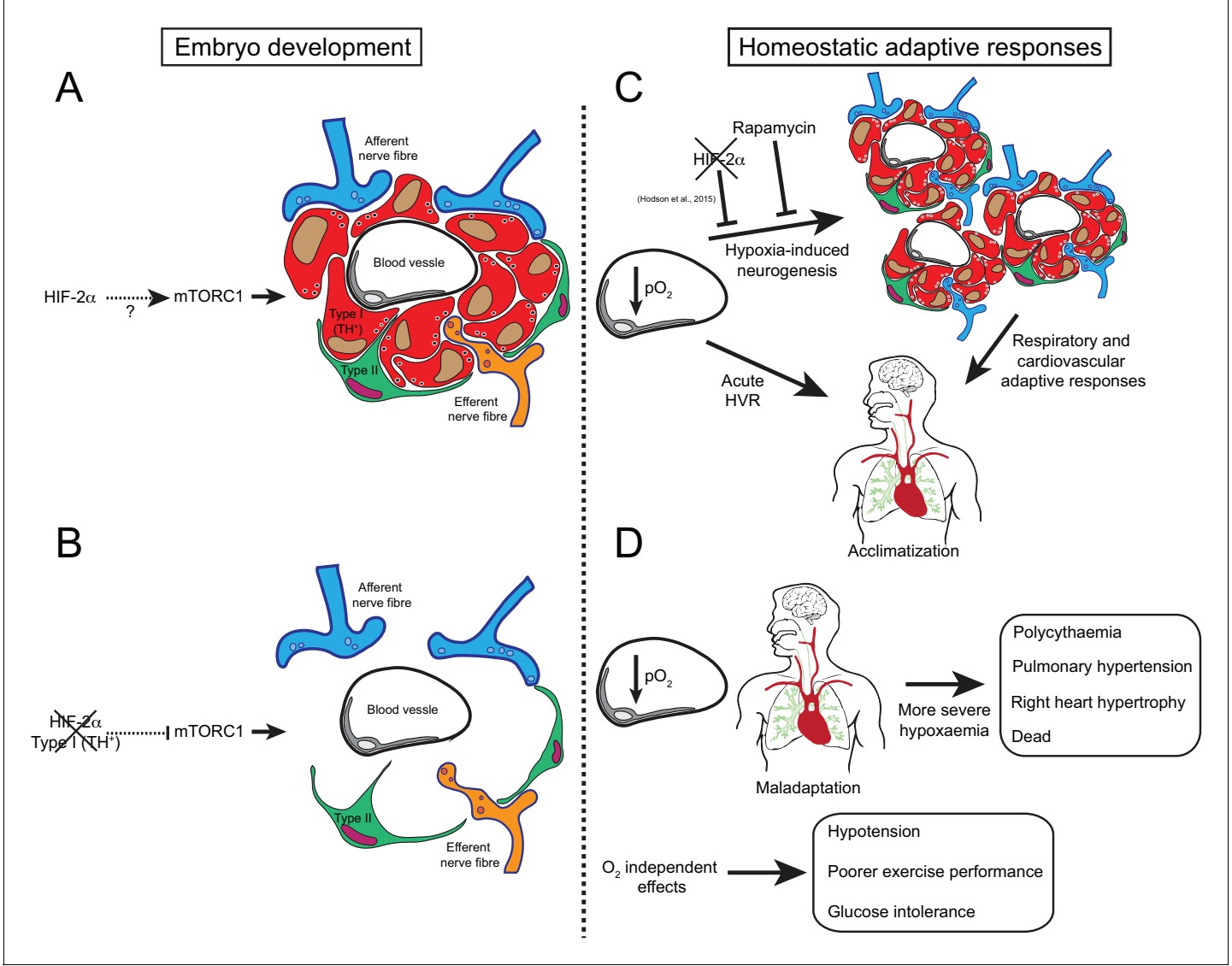

**Figure 8.** Graphic abstract of the role of HIF-2α in CB development and function. (**A**) Schematic representation of a CB glomerulus. In CB development mTORC1 activity is high, which likely contributes to normal CB proliferation and growth. (**B**) Embryonic *Epas1 (Hif-2α)* ablation in TH⁺ cells (type I cells) leads to repressed mTORC1 activity and progressive O₂-sensitive cell loss via a direct or indirect mechanism. (**C**) Upon low arterial pO₂, CB O₂-sensitive glomus cells trigger an acute hypoxic ventilatory response (HVR) increasing O₂ uptake by the lungs. Sustained hypoxia induces CB growth, a process that is HIF-2α-dependent (*Hodson et al., 2016*) and blocked by rapamycin (mTORC1 inhibitor). Hypoxia-induced CB neurogenesis further increases ventilation and regulates cardiovascular function allowing adaptation. (**D**) The absence of CB responses to low pO₂ leads to a more severe hypoxaemia, resulting in pathological polycythaemia, pulmonary hypertension, and right ventricular failure. Other adaptive responses, such as arterial pressure, exercise performance and glucose homeostasis, are also impaired by CB loss. Taken together, we show that HIF-2α is essential for CB development and that its absence results in the disruption of key homeostatic adaptive responses.

DOI: https://doi.org/10.7554/eLife.34681.017

chronic obstructive pulmonary disease, sudden death infant syndrome and hypertension (*Iturriaga et al., 2016*; *López-Barneo et al., 2016b*).

We describe here the first mouse model with specific loss of carotid body O₂-sensitive cells: a model that is viable in normoxia but shows impaired physiological adaptations to both acute and chronic hypoxia. In addition, we describe new and likely important roles for the CB in systemic blood pressure regulation, exercise performance and glucose management (*Figure 8*). Further investigation of this mouse model could aid in revealing roles of the carotid body in both physiology and disease.

# Materials and methods

## Key resources table

| Reagent type (species) or resource | Designation | Source or reference | Identifiers | Additional information |
|---|---|---|---|---|
| Gene (*Mus musculus*) | *Hif1a* | NA | MGI:106918 | |
| Gene (*M. musculus*) | *Epas*1 | NA | MGI:109169 | |
| Gene (*M. musculus*) | *Th* | NA | MGI:98735 | |
| Strain, strain background (*M. musculus*) | C57BL/6J | The Jackson laboratory | Jax:000664 | |
| Genetic reagent (*M. musculus*) | HIF-1α$^{WT}$ | PMID:10945599 | Jax: 007561 | |
| Genetic reagent (*M. musculus*) | HIF-2α$^{WT}$ | PMID:17284606 | Jax:008407 | |
| Genetic reagent (*M. musculus*) | TH-Cre | PMID:15452869 | MGI:3056580 | |
| Genetic reagent (*M. musculus*) | Td-Tomato | PMID:20023653 | Jax:007909 | |
| Antibody | anti-TH (rabbit polyclonal) | Novus Biologicals | Novus Biologicals :NB300-109 | (1:1000) |
| Antibody | anti-TH (chicken polyclonal) | abcam | abcam:ab76442 | (1:1000) |
| Antibody | anti-GFAP (rabbit polyclonal) | Dako | Dako:Z0334 | (1:500) |
| Antibody | anti-BrdU (rat polyclonal) | abcam | abcam:ab6326 | (1:100) |
| Antibody | anti-phospho S6 (rabbit polyclonal) | Cell Signaling Technology | Cell signaling:2211 | (1:200) |
| Antibody | anti-SMAα(mouse monoclonal) | Sigma-Aldrich | Sigma-Aldrich:A-2547 | (1:200) |
| Antibody | Alexa 488- or 568- secondaries | ThermoFisher | | (1:500) |
| Sequence-based reagent | *Hif1a*_probe | IDT | | 5'-/56-FAM/CCTGTTGGTTGCGC AGCAAGCATT/3BHQ_1/–3' |
| Sequence-based reagent | *Hif1a*_F | IDT | | 5'-GGTGCTGGTGTCCAAAATGTAG-3' |
| Sequence-based reagent | *Hif1a*_R | IDT | | 5'-ATGGGTCTAGAGAGAT AGCTCCACA-3' |
| Sequence-based reagent | *Epas1*_probe | IDT | | 5'-/56-FAM/CCACCTGGA/ZEN/ CAAAGCCTCCATCAT/3IABkFQ/−3' |
| Sequence-based reagent | *Epas1*_F | IDT | | 5'-TCTATGAGTTGGCTCATGAGTTG-3' |
| Sequence-based reagent | *Epas1*_R | IDT | | 5'-GTCCGAAGGAAGCTGATGG-3' |
| Peptide, recombinant protein | AngII | Sigma-Aldrich | Sigma-Aldrich:A2900 | |
| Peptide, recombinant protein | Insulin (human) | Sigma-Aldrich | Sigma-Aldrich:11 376 497 001 | |
| Peptide, recombinant protein | Collagenase type IA | Sigma-Aldrich | Sigma-Aldrich:C9891 | |
| Peptide, recombinant protein | Trypsin | Sigma-Aldrich | Sigma-Aldrich:93610 | |
| Commercial assay or kit | LSAB+, Dako REAL Detection Systems | Dako | Dako:K5001 | |

*Continued on next page*

*Continued*

| Reagent type (species) or resource | Designation | Source or reference | Identifiers | Additional information |
|---|---|---|---|---|
| Commercial assay or kit | Epinephrine/Norepinephrine ELISA Kit | Abnova | Abnova:KA1877 | |
| Commercial assay or kit | DNeasy Blood and Tissue Kit | Qiagen | Qiagen:69506 | |
| Commercial assay or kit | TaqMan Universal PCR Master Mix | Applied Biosystems | Applied biosystems:4304437 | |
| Commercial assay or kit | Click-iT TUNEL Alexa Fluor 488 Imaging Assay | ThermoFisher | ThermoFisher:C10245 | |
| Chemical compound, drug | BrdU | Sigma-Aldrich | Sigma-Aldrich:B5002 | |
| Chemical compound, drug | Rapamycin | LC Laboratories | LC Laboratories:R-5000 | |
| Chemical compound, drug | D-Glucose | Sigma-Aldrich | Sigma-Aldrich:G8270 | |
| Chemical compound, drug | Sodium L-Lactate | Sigma-Aldrich | Sigma-Aldrich:71718 | |
| Software, algorithm | Fiji | | | http://imagej.net/Fiji |
| Other | EVG stain | Merck | Merck:115974 | |
| Other | Hematoxylin Solution, Mayer's | Sigma-Aldrich | Sigma-Aldrich:MHS16 | |
| Other | Eosin Y solution, aqueous | Sigma-Aldrich | Sigma-Aldrich:HT110216 | |
| Other | DAPI stain | Molecular probes | Molecular probes:D1306 | (1:5000) |

## Mouse husbandry

This research has been regulated under the Animals (Scientific Procedures) Act 1986 Amendment Regulations 2012 following ethical review by the University of Cambridge Animal Welfare and Ethical Review Body (AWERB) Home Office Licenses 80/2618 and PC64B8953. Targeted deletion of *Hif1a*, *Epas1* and *Td-Tomato* in sympathoadrenal tissues was achieved by crossing homozygous mice carrying loxP-flanked alleles *Hif1a$^{tm3Rsjo}$* (**Ryan et al., 1998**), *Epas1$^{tm1Mcs}$* (**Gruber et al., 2007**) or *Gt (ROSA)26Sor$^{tm9(CAG-tdTomato)Hze}$* (**Madisen et al., 2010**) into a mouse strain of *cre* recombinase expression driven by the endogenous tyrosine hydroxylase promoter (*Th$^{tm1(cre)Te}$*)(**Lindeberg et al., 2004**), which is specific of catecholaminergic cells. LoxP-flanked littermate mice without cre recombinase (*Hif1a$^{flox/flox}$*, *Epas1$^{flox/flox}$* or *Td-Tomato$^{flox/flox}$*) were used as control group (HIF-1α$^{WT}$, HIF-2α$^{WT}$ and Td-Tomato, respectively) for all the experiments. C57BL/6J wild type mice were purchased from The Jackson laboratory (Bar Harbor, Maine, US). All the experiments were performed with age and sex matched control and experimental mice.

## Tissue preparation, immunohistochemistry and morphometry

Newborn (P0) and adult mice (8–12 weeks old) were killed by an overdose of sodium pentobarbital injected intraperitoneally. E18.5 embryos were collected and submerged into a cold formalin solution (Sigma-Aldrich, Dorset, UK). The carotid bifurcation and adrenal gland, were dissected, fixed for 2 hr with 4% paraformaldehyde (Santa Cruz Biotechnology) and cryopreserved (30% sucrose in PBS) for cryosectioning (10 μm thick, Bright Instruments, Luton, UK). Tyrosine hydroxylase (TH), glial fibrillary acidic protein (GFAP), bromodeoxyuridine (BrdU) and phospho-S6 ribosomal protein (p-S6) were detected using the following polyclonal antibodies: rabbit anti-TH (1:5000, Novus Biologicals, Abindong, UK; NB300-109) for single immunofluorescence or chicken anti-TH (1:1000, abcam, Cambridge, UK; ab76442) for double immunofluorescence, rabbit anti-GFAP (1:500, Dako, Cambridge, UK; Z0334), rat anti-BrdU (1:100, abcam, ab6326), and rabbit anti-phospho-S6 ribosomal protein (1:200, Cell Signaling, 2211), respectively. Fluorescence-based detection was visualized using, Alexa-Fluor 488- and 568-conjugated anti-rabbit IgG (1:500, ThermoFisher, Waltham, MA US), Alexa-Fluor 488-conjugated anti-chicken IgG (1:500, ThermoFisher) and Alexa-Fluor 488-conjudaged anti-rat IgG

(1:500, ThermoFisher) secondary antibodies. Chromogen-based detection was performed with Dako REAL Detection Systems, Peroxidase/DAB+, Rabbit/Mouse (Dako, K5001) and counterstained with Carrazzi hematoxylin. For the Td-Tomato detection, carotid bifurcation and adrenal gland were dissected, embedded with OCT, flash-frozen into liquid $N_2$ and sectioned (10 μm thick, Bright Instruments). Direct Td-Tomato fluorescence was imaged after nuclear counterstain with 4',6-diamidino-2-phenylindole (DAPI).

CB glomus cell counting, area measurements and pixel mean intensity were calculated on micrographs (Leica DM-RB, Wetzlar, Gernany) taken from sections spaced 40 μm apart across the entire CB using Fiji software (*Schindelin et al., 2012*) as previously reported (*Macías et al., 2014*). CB and SCG volume was estimated on the same micrographs according to Cavalieri's principle. Lung tissue was removed in the distended state by infusion into the trachea of 0.8% agarose and then fixed in 4% paraformaldehyde for paraffin embedding. Lung sections (6 μm thick) were stained with H and E and EVG stain to assess morphology (Merck, Darmstadt, Germany). Vessel medial thickness was measured on micrographs using Fiji software. To determine the degree of muscularization of small pulmonary arteries, serial lung tissue sections were stained with anti-α smooth muscle actin (1:200, Sigma-Aldrich, A2547). Heart and spleen sections (6 μm thick) were stained with H and E to assess morphology. Tissue sections were independently coded and quantified by a blinded investigator.

## Gene deletion

Genomic DNA was isolated from dissected SCG and AM using DNeasy Blood and Tissue Kit (Qiagen, Hilden, Germany). Relative gene deletion was assayed by qPCR (One-Step Plus Real-Time PCR System; ThermoFisher Scientific) using TaqMan Universal PCR Master Mix (ThermoFisher Scientific) and PrimeTime Std qPCR Assay (IDT, Coralville, IA US).

For *Hif1a*,
*Hif1a*_probe: 5'-/56-FAM/CCTGTTGGTTGCGCAGCAAGCATT/3BHQ_1/–3'
*Hif1a*_F: 5'-GGTGCTGGTGTCCAAAATGTAG-3'
*Hif1a*_R: 5'-ATGGGTCTAGAGAGATAGCTCCACA-3'
For *Epas1*,
*Epas1*_probe: 5'-/56-FAM/CCACCTGGACAAAGCCTCCATCAT/3IABkFQ/−3'
*Epas1*_F: 5'-TCTATGAGTTGGCTCATGAGTTG-3'
*Epas1*_R: 5'-GTCCGAAGGAAGCTGATGG-3'
Relative gene deletion levels were calculated using the 2ΔΔCT method.

## Tissue preparation for amperometry and patch clamp experiments

Adrenal glands and SCG were quickly removed and transferred to ice-cooled PBS. Dissected AM was enzymatically dispersed in two steps. First, tissue was placed in 3 ml of extraction solution (in mM: 154 NaCl, 5.6 KCl, 10 Hepes, 11 glucose; pH 7.4) containing 425 U/ml collagenase type IA and 4–5 mg of bovine serum albumin (all from Sigma-Aldrich), and incubated for 30 min at 37°C. Tissue was mechanically dissociated every 10 min by pipetting. Then, cell suspension was centrifuged (165 g and 15–20°C for 4 min), and the pellet suspended in 3 ml of PBS containing 9550 U/ml of trypsin and 425 U/ml of collagenase type IA (all from Sigma-Aldrich). Cell suspension was further incubated at 37°C for 5 min and mechanically dissociated by pipetting. Enzymatic reaction was stopped by adding ≈10 ml of cold complete culture medium (DMEM/F-12 (21331–020 GIBCO, ThermoFisher Scientific) supplemented with penicillin (100 U/ml)/streptomycin (10 μg/ml), 2 mM L-glutamine and 10% fetal bovine serum. The suspension was centrifuged (165 g and 15–20°C for 4 min), the pellet gently suspended in 200 μl of complete culture medium. Next, cells were plated on small coverslips coated with 1 mg/ml poly-L-lysine and kept at 37°C in a 5% $CO_2$ incubator for settlement. Afterwards, the petri dish was carefully filled with 2 ml of culture medium and returned to the incubator for 2 hr. Under these conditions, cells could be maintained up to 12–14 hr to perform experiments.

To obtain SCG dispersed cells, the tissue was placed in a 1.5 ml tube containing enzymatic solution (2.5 mg/ml collagenase type IA in PBS) and incubated for 15 min in a thermoblock at 37°C and 300 rpm. The suspension was mechanically dissociated by pipetting and centrifuged for 3 min at 200 g. The pellet was suspended in the second enzymatic dispersion (1 mg/ml of trypsin in PBS), incubated at 37°C, 300 rpm for 25 min and dissociated again with a pipette. Enzymatic reaction was stopped adding a similar solution than used for chromaffin cells (changing fetal bovine serum for

newborn calf serum). Finally, dispersed SCG cells were plated onto poly-L-lysine coated coverslips and kept at the 37°C incubator. Experiments with SCG cells were done between 24–48 hr after finishing the culture.

To prepare AM slices, the capsule and adrenal gland cortex was removed and the AM was embedded into low melting point agarose at when reached 47°C. The agarose block is sectioned (200 μm thick) in an $O_2$-saturated Tyrode's solution using a vibratome (VT1000S, Leica). AM slices were washed twice with the recording solution (in mM: 117 NaCl, 4.5 KCl, 23 NaHCO$_3$, 1 MgCl$_2$, 1 CaCl$_2$, 5 glucose and five sucrose) and bubbled with carbogen (5% CO$_2$ and 95% O$_2$) at 37°C for 30 min in a water bath. Fresh slices are used for the experiments during 4–5 hr after preparation.

## Patch-clamp recordings

The macroscopic currents were recorded from dispersed mouse AM and SCG cells using the whole cell configurations of the patch clamp technique. Micropipettes ($\approx 3$ MΩ) were pulled from capillary glass tubes (Kimax, Kimble Products, Rockwood, TN US) with a horizontal pipette puller (Sutter instruments model P-1000, Novato, CA US) and fire polished with a microforge MF-830 (Narishige, Amityville, NY US). Voltage-clamp recordings were obtained with an EPC-7 amplifier (HEKA Electronik, Lambrecht/Pfalz, Germany). The signal was filtered (3 kHz), subsequently digitized with an analog/digital converter (ITC-16 Instrutech Corporation, HEKA Electronik) and finally sent to the computer. Data acquisition and storage was done using the Pulse/pulsefit software (Heka Electronik) at a sampling interval of 10 μsec. Data analysis is performed with the Igor Pro Carbon (Wavemetrics, Portland, OR US) and Pulse/pulsefit (HEKA Electronik) programs. All experiments were conducted at 30–33°C. Macroscopic Ca$^{2+}$, Na$^+$, and K$^+$ currents were recorded in dialyzed cells. The solutions used for the recording of Na$^+$ and Ca$^{2+}$ currents contained (in mM): external: 140 NaCl, 10 BaCl$_2$, 2.5 KCl, 10 HEPES, and 10 glucose; pH 7.4 and osmolality 300 mOsm/kg; and internal: 110 CsCl, 30 CsF, 10 EGTA, 10 HEPES, and 4 ATP-Mg; pH 7.2 and osmolality 285 mOsm/kg. The external solution used for the recording of whole cell K$^+$ currents was (in mM): 117 NaCl, 4.5 KCl, 23 NaHCO$_3$, 1 MgCl$_2$, 2.5 CaCl$_2$, 5 glucose and five sucrose. The internal solution to record K$^+$ currents was (in mM): 80 potassium glutamate, 50 KCl, 1 MgCl$_2$, 10 HEPES, 4 MgATP, and 5 EGTA, pH 7.2. Depolarizing steps of variable duration (normally 10 or 50 msec) from a holding potential of −70 mV to different voltages ranging from −60 up to +50 mV (with voltage increments of 10 mV) were used to record macroscopic ionic currents in AM and SCG cells.

## Amperometric recording

To perform the experiments, single slices were transferred to the chamber placed on the stage of an upright microscope (Axioscope, Zeiss, Oberkochen, Germany) and continuously perfused by gravity (flow 1–2 ml min$^{-1}$) with a solution containing (in mM): 117 NaCl, 4.5 KCl, 23 NaHCO$_3$, 1 MgCl$_2$, 2.5 CaCl$_2$, 5 glucose and five sucrose. The 'normoxic' solution was bubbled with 5% CO$_2$, 20% O$_2$ and 75 % N$_2$ (PO$_2 \approx$ 150 mmHg). The 'hypoxic' solution was bubbled with 5% CO$_2$ and 95% N$_2$ (PO$_2 \approx$ 10–20 mmHg). The 'hypercapnic' solution was bubbled with 20% CO$_2$, 20% O$_2$ and 60 % N$_2$ (PO$_2 \approx$ 150 mmHg). The osmolality of solutions was $\approx$ 300 mosmol kg$^{-1}$ and pH = 7.4. In the 20 mM K$^+$ solution, NaCl was replaced by KCl equimolarly (20 mM). All experiments were carried out at a chamber temperature of 36°C. Amperometric currents were recorded with an EPC-8 patch-clamp amplifier (HEKA Electronics), filtered at 100 Hz and digitized at 250 Hz for storage in a computer. Data acquisition and analysis were carried out with an ITC-16 interface and PULSE/PULSEFIT software (HEKA Electronics). The secretion rate (fC min$^{-1}$) was calculated as the amount of charge transferred to the recording electrode during a given period of time.

## Catecholamine measurements

Catecholamine levels were determined by ELISA (Abnova, Taipei, Taiwan) following manufacture's instructions. Spontaneous urination was collected from control and experimental mice (8–12 weeks old) at similar daytime (9:00am). HIF-2α$^{WT}$and TH-HIF-2α$^{KO}$ or HIF-1α$^{WT}$, TH-HIF-1α$^{KO}$ samples were collected and assayed in different days.

## Tunel assay

Cell death was determined on CB sections using Click-iT TUNEL Alexa Fluor 488 Imaging Assay (ThermoFisher Scientific), following manufacture's indications. TUNEL$^+$ cells across the whole CB parenchyma were counted and normalized by area.

## Plethysmography

Respiratory parameters were measured by unrestrained whole-body plethysmography (Data Sciences International, St. Paul, MN US). Mice were maintained in a hermetic chamber with controlled normoxic airflow (1.1 l/min) of 21% $O_2$ ($N_2$ balanced, 5000ppm $CO_2$, BOC Healthcare, Manchester, UK) until they were calm (30–40 min). Mice were then exposed to pre-mixed hypoxic air (1.1 l/min) of 10% $O_2$ ($N_2$ balanced, 5000ppm $CO_2$, BOC Healthcare) or 5% $CO_2$ (21% $O_2$, $N_2$ balanced, BOC, Healthcare) for 30 min (time-course experiments) or 10 min (VAH and hypercapnia experiments). Real-time data were recorded and analyzed using Ponemah software (Data Sciences International). Each mouse was monitored throughout the experiment and periods of movements and/or grooming were noted and subsequently removed from the analysis. The transition periods between normoxia and hypoxia (2–3 min) were also removed from the analysis.

## Oxygen saturation

Oxygen saturation was measured using MouseOx Pulse Oximeter (Starr Life Sciences Corp, Oakmont, PA US) on anaesthetised (isofluorane) mice. Anaesthetic was vaporized with 2 l/min of 100% $O_2$ gas flow (BOC Healthcare) and then shift to 2 l/min of pre-mixed 10% $O_2$ flow ($N_2$ balanced, 5000ppm $CO_2$, BOC Healthcare) for 5 min. Subsequently, gas flow was switched back to 100% $O_2$. For oxygen saturation in normoxia anaesthetic was vaporized with 2 l/min of 21% $O_2$ gas flow.

## Whole-body metabolic assessments

Energy expenditure was measured and recorded using the Columbus Instruments Oxymax system (Columbus, OH US) according to the manufacturer's instructions. Mice were randomly allocated to the chambers and they had free access to food and water throughout the experiment with the exception of fasting experiments. An initial 18–24 hr acclimation period was disregarded for all the experiments. Once mice were acclimated to the chamber, the composition of the influx gas was switched from 21% $O_2$ to 10% $O_2$ using a PEGAS mixer (Columbus Instruments). For maximum exertion testing, mice were allowed to acclimatize to the enclosed treadmill environment for 15 min before running. The treadmill was initiated at five meters/min and the mice warmed up for 5 min. The speed was increased up to 27 meters/min or exhaustion at 2.5 min intervals on a 15° incline. Exhaustion was defined as the third time the mouse refused to run for more than 3–5 s or when was unable to stay on the treadmill. Blood glucose and lactate levels were measured before and after exertion using a StatStrip Xpress (nova biomedical, Street-Waltham, MA US) glucose and lactate meters.

## Chronic hypoxia exposure and in vivo CB proliferation

Mice were exposed to a normobaric 10% $O_2$ atmosphere in an enclosed chamber with controlled $O_2$ and $CO_2$ levels for up to 21 days. Humidity and $CO_2$ levels were quenched throughout the experiments by using silica gel (Sigma-Aldrich) and spherasorb soda lime (Intersurgical, Wokingham, UK), respectively. Haematocrit was measured in a bench-top haematocrit centrifuge. Haemoglobin levels and blood cell counts were analyzed with a Vet abc analyzer (Horiba, Kyoto, Japan) according to the manufacture's instructions. Heart, spleen tissues were removed and their wet weights measured. Right ventricular systolic pressures were assessed using a pressure-volume loop system (Millar, Houston, TX US) as reported before (*Cowburn et al., 2016*). Mice were anaesthetized (isoflurane) and right-sided heart catheterization was performed through the right jugular vein. Lungs were processed for histology as describe above. Analogous procedures were followed for mice maintained in normoxia.

For hypoxia-induced in vivo CB proliferation studies, mice were injected intraperitoneally once a week with 50 mg/kg of BrdU (Sigma-Aldrich) and constantly supplied in their drinking water (1 mg/ml) as previously shown (*Pardal et al., 2007*). In vivo mTORC inhibition was achieved by intraperitoneal injection of rapamycin (6 mg kg$^{-1}$ day$^{-1}$). Tissues were collected for histology after 14 days at

10% $O_2$ as described above. VAH was recorded in vehicle control and rapamycin-injected C57BL/6J mice after 7 days in chronic hypoxia by plethysmography as explained above.

## Blood pressure measurement by radio-telemetry

All radio-telemetry hardware and software was purchased from Data Science International. Surgical implantation of radio-telemetry device was performed following University of Cambridge Animal Welfare and Ethical Review Body's (AWERB) recommendations and according to manufacturer's instructions. After surgery, mice were allowed to recover for at least 10 days. All baseline telemetry data were collected over a 72 hr period in a designated quiet room to ensure accurate and repeatable results. Blood pressure monitoring during hypoxia challenge was conducted in combination with Columbus Instruments Oxymax system and PEGAS mixer. Mice were placed in metabolic chambers and allowed to acclimate for 24 hr before the oxygen content of the flow gas was reduced to 10%. Mice were kept for 48 hr at 10% $O_2$ before being returned to normal atmospheric oxygen for a further 24 hr. For hypertension studies, mice were subsequently implanted with a micro osmotic pump (Alzet, Cupertino, CA; 1002 model) for Ang II (Sigma-Aldrich) infusion (490 ng $min^{-1}$ $kg^{-1}$) for 2 weeks. Blood pressure, heart rate, temperature and activity were monitored for a 48 hr period after 7, 14 and 21 days after osmotic pump implantation.

## Glucose, insulin and lactate tolerant tests

All the blood glucose and lactate measurements were carried out with a StatStrip Xpress (nova biomedical) glucose and lactate meters. Blood was sampled from the tail vein. For glucose tolerance test (GTT), mice were placed in clean cages and fasted for 14–15 hr (overnight) with free access to water. Baseline glucose level (0 time point) was measured and then a glucose bolus (Sigma-Aldrich) of 2 g/kg in PBS was intraperitoneally injected. Glucose was monitored after 15, 30, 60, 90, 120 and 150 min. For insulin tolerance test, mice were fasted for 4 hr and baseline glucose was measured (0 time point). Next, insulin (Roche, Penzberg, Germany) was intraperitoneally injected (0.75 U/kg) and blood glucose content recorded after 15, 30, 60, 90 and 120 min. For lactate tolerance test, mice were injected with a bolus of 2 g/kg of sodium lactate (Sigma-Aldrich) in PBS and blood lactate measured after 15, 30, 60, 90 and 120 min.

## Statistical analysis

Particular statistic tests and sample size can be found in the figures and figure legends. Data are presented as the mean ± standard error of the mean (SEM). Shapiro-Wilk normality test was used to determine Gaussian data distribution. Statistical significance between two groups was assessed by un-paired *t*-test in cases of normal distribution with a Welch's correction if standard deviations were different. In cases of non-normal distribution (or low sample size for normality test) the non-parametric Mann-Whitney U test was used. For multiple comparisons we used two-way ANOVA followed by Fisher's LSD test. Non-linear regression (one-phase association) curve fitting, using a least squares method, was used to model the time course data in *Figures 3* and *5* and *Figure 6—figure supplement 1*. Extra sum-of-squares F test was used to compare fittings, and determine whether the data were best represented by a single curve, or by separate ones. Individual area under the curve (AUC) was calculated for the circadian time-courses in *Figures 3* and *5*; and *Figure 6—figure supplement 1* and *Figure 7—figure supplement 1*, followed by un-paired t-test or non-parametric Mann-Whitney U test as explained above. Statistical significance was considered if p≤0.05. Analysis was carried out using Prism6.0f (GraphPad software, La Jolla, CA US) for Mac OS X.

## Acknowledgements

This work is funded by the Wellcome Trust (grant WT092738MA).

## Additional information

### Funding

| Funder | Grant reference number | Author |
|---|---|---|
| Wellcome | WT092738 | Randall Johnson |

The funders had no role in study design, data collection and interpretation, or the decision to submit the work for publication.

### Author contributions

David Macias, Conceptualization, Data curation, Formal analysis, Validation, Investigation, Methodology, Writing—original draft, Writing—review and editing; Andrew S Cowburn, Formal analysis, Investigation, Methodology, Writing—review and editing; Hortensia Torres-Torrelo, Data curation, Investigation; Patricia Ortega-Sáenz, Investigation, Methodology; José López-Barneo, Data curation, Formal analysis, Supervision, Methodology, Writing—review and editing; Randall S Johnson, Conceptualization, Formal analysis, Supervision, Funding acquisition, Writing—original draft, Project administration, Writing—review and editing

### Author ORCIDs

David Macias ⓘ http://orcid.org/0000-0002-8676-1964

Randall S Johnson ⓘ http://orcid.org/0000-0002-4084-6639

### Ethics

Animal experimentation: This work was carried out with approval and following review of the University of Cambridge AWERB (Animal Welfare Ethical Review Board) and under license of the UK Home Office (Home Office License numbers 80/2618 and PC64B8953).

### Decision letter and Author response

Decision letter https://doi.org/10.7554/eLife.34681.022

Author response https://doi.org/10.7554/eLife.34681.023

## Additional files

### Supplementary files

• Transparent reporting form

DOI: https://doi.org/10.7554/eLife.34681.018

### Data availability

All data generated or analysed during this study are included in the manuscript and supporting files.

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
