## [Decision Letter]

Thank you for submitting your article "HIF-2α is essential for carotid body development and function" for consideration by *eLife*. Your article has been reviewed by three peer reviewers, and the evaluation has been overseen by a Reviewing Editor and Marianne Bronner as the Senior Editor. The following individuals involved in review of your submission have agreed to reveal their identity: Peter Ratcliffe (Reviewer #1).

The reviewers have discussed the reviews with one another and the Reviewing Editor has drafted this decision to help you prepare a revised submission.

Macias et al. describe novel mouse lines that lack specifically HIF1α of HIF2α in tyrosine hydroxylase (TH) expressing cells, and show using these mice that HIF2α is an absolute requirement for the growth and development of the oxygen sensing glomus cells of the carotid body (CB) increased apoptosis being the reason for the lack of CB cell loss. TH-HIF2α^KO^ mice have impaired ventilation response to hypoxia, decreased survival under hypoxia, increased erythropoietic response to hypoxia and extramedullary erythropoiesis. Moreover, their blood pressure (BP) is impaired at baseline and in response to hypoxia and they are susceptible to a severe BP impairment in an angiotensin-induced hypertension model. Yet, these mice show some alteration in their VO_2_ max during exercise and in their glucose tolerance. The authors identify defective mTORC1 activation as the reason for defective CB development.

All three reviewers found the data interesting, and although critical comments were raised, no major problems were identified. The overall conclusion based on the reviews was to encourage submission of a revised version, where the very detailed comments of the reviewers have been addressed. In particular, several suggestions were made to clarify the conclusions based on the massive data, and on their relationship with previously published data. A schematic figure illustrating the major conclusions would be an excellent addition to the revised manuscript.

In the discussion after the reviews were completed, reviewer #1 offered the following specific advice:

I agree with reviewer 2 that a number of the findings lack a clear mechanism but am comfortable as long as this is clearly pointed out. More specifically it is important to distinguish as far as is possible which/whether aspects of the systemic phenotype derive indirectly from disturbances of systemic oxygenation as opposed to carotid body function itself and whether (as pointed out by reviewer) recombination in other organs might contribute to the phenotypes. If, after reasonable efforts, this is not possible then it is important that such caveats to the data interpretation are made overt.

*Reviewer #1:*

This paper describes a critical role for the integrity of HIF-2α, but not HIF-1α in the development of the carotid body. In mice where HIF-α genes are inactivated by TH-driven expression of Cre, the adrenal medulla and superior cervical ganglion apparently develop and function normally, whereas the carotid body is vestigial. The authors go on to report on a wide range of physiological testing in the mice, which have essentially non-functional carotid bodies.

As an entry to mechanistic analysis of the role of HIF-2α in carotid body development, they also show that in pre-natal (18.5dpc) mice there is reduced staining of phosphorylated ribosomal protein S6, which is expressed strongly at this stage in wild-type embryos. It is also shown that during adaptation to hypoxia in the adult mouse (when a proliferative enlargement of the carotid body is seen) there is also an increase in phospho-S6 staining, which is blocked by rapamycin. This is associated with reduced proliferation of TH^+^ (type 1) cells.

Given the importance of the carotid body to the physiology of hypoxia, these findings are of interest. Some points however require clarification.

i) As the authors outline (Discussion, first paragraph), the original descriptions of constitutive HIF-α inactivation phenotypes included the association of HIF-2α inactivation with failure of sympatho-adrenal development. The authors then state that the current findings argue against a 'global role for the (HIF-2α) gene in development of catecholamine tissues'. The most obvious difference between this and the earlier reports is the use of TH-IRES-Cre 'driver' versus constitutive inactivation. What do the authors know of the timing of Cre activation using this strain of mouse? Is it possible that the TH promoter used in the driver-strain only becomes active during later stages of sympatho-adrenal development, and that the current (carotid body – restricted) phenotype reflects this i.e. a more extensive phenotype might have been generated if the deletion of HIF-2α had occurred earlier in the sympatho-adrenal lineage? Further discussion of this is required to set the work in the context of that earlier literature.

ii) The authors describe extensive (and interesting) physiological differences between wild type mice and the TH-HIF-2α^KO^ animals. Some of these differences occur in normoxic animals and some (e.g. pulmonary vascular responses) are seen only in response to hypoxia. If possible it would be important to distinguish those effects which are directly due to abnormal carotid body function (i.e. altered neurogenic responses) from those are secondary to increased hypoxia. For example, it is stated that 'pulmonary adaption to chronic hypoxia is highly reliant on a normally functioning CB'. For the general reader it would be important to make it clear that such a connection may well be indirect, (due to increased hypoxia) rather implying a direct role of the carotid body. Along similar lines, what is known of the oxygen saturation of the TH-HIF-2α^KO^ animals in normoxia? Are they always fully saturated – or do they manifest desaturation at any stage? Even transient desaturation could have important effects on the parameters that are being described. Ideally data on this, or at least clear discussion as to whether this has or has not been excluded, is important.

iii) In the first paragraph of the subsection “CB development and proliferation in response to hypoxia require mTORC1 activation” the authors argue that hypoxia is an mTORC1 activator. Greater clarity on both the background literature and interpretation of the current work is required – particularly as hypoxia can also be an mTORC1 inhibitor due to metabolic effects. Specifically it would be helpful if the authors could discuss briefly what connections between HIF-2α and mTORC1 have been established previously and what (direct or indirect connections) they are proposing on the basis of the current work.

iv) Although early work apparently implicated HIF-1α as a key factor in carotid body regulation, the current work builds on a body on more recent data indicating the importance of HIF-2α rather than HIF-1α in carotid body physiology. In the Introduction the authors state 'mouse models hemizygous for HIFs-α or where HIF inactivation was induced in adults showed contrasting effects on CB function without in either case affecting CB morphogenesis'. In fact the paper by Hodson et al. examined both hemizygous animals and biallelic inactivation in the adult. It reported that in both settings HIF-2α (not HIF-1α) inactivation had major effects on functional responses to sustained hypoxia and that the proliferative (i.e. morphological) response of the carotid bodies to sustained hypoxia was entirely ablated after HIF-2α but not HIF-1α^KO^. These findings are closely relevant to (and consistent with) the current work and should be described more clearly in the introduction. Notwithstanding this, I do consider that the current work is an important advance as outlined above.

v) Subsection “Sympathoadrenal Hif-2α loss blocks carotid body glomus cell development”, last paragraph. I am not sure that the greater number of TUNEL+ cells in HIF-2α^KO^ animals indicates an essential function in differentiation (as opposed to survival). Could the authors clarify or modify?

vi) Can the authors comment on potential reasons why the hypoxic HIF-2α^KO^ animals suffered haemorrhages?

vii) Figure 1—figure supplement 1A please clarify designation of upper and lower panels i.e. TH-Cre = (activated tDTomato).

viii) Figure 3—figure supplement 1 – is there a greater response to CO_2_ in the TH-HID-2α adrenal medulla or is this appearance due to different scales – could the authors comment?

*Reviewer #2:*

Macias et al. describe novel mouse lines that lack specifically HIF1α of HIF2α in tyrosine hydroxylase (TH) expressing cells, and show using these mice that HIF2α is an absolute requirement for the growth and development of the oxygen sensing glomus cells of the carotid body (CB) increased apoptosis being the reason for the lack of CB cell loss. TH-HIF2α^KO^ mice, unlike TH-HIF1α^KO^ mice, have impaired ventilation response to hypoxia, decreased survival under hypoxia, increased erythropoietic response to hypoxia and extramedullary erythropoiesis. Moreover, their blood pressure (BP) is impaired at baseline and in response to hypoxia and they are susceptible to a severe BP impairment in an angiotensin-induced hypertension model. Yet, these mice show some alteration in their VO_2_ max during exercise and in their glucose tolerance. Finally, the authors identify defective mTORC1 activation as the reason for defective CB development.

The manuscript contains a very large amount of high quality data from demanding experimental set ups and is well written. However, it lacks coherence and a clear conclusion. The amount of data provided is overwhelming and some reduction should be considered to make the manuscript more coherent. The manuscript would certainly benefit from a conclusive figure (schematic).

I have several concerns that warrant my enthusiasm for the publication of the manuscript.

1) What is the deletion efficiency of HIF1α and HIF2α in the corresponding CB KO cells? In adult TH-HIF2α^KO^s it may not be measurable but at P0 or E18.5 CB cells are still available for HIF2α^KO^s.

2) The conclusion that mTORC1 activation is required for CB expansion in a HIF2α-dependent manner cannot be drawn from the experiments performed. In order to draw such a conclusion for example the rapamycin experiment should be repeated with TH-HIF2α^KO^s. Regarding the mechanism, did rapamycin induce apoptosis of CB cells?

3) As both, VO_2_ and VCO_2_ change after hypoxia (Figure 3J and K) – please show RER.

4) In the Angiotensin II-induced hypertension model the difference in TH-HIF2α^KO^ BP is only seen at 7 day (not at 14 or 21 days). What might be the reasons for this?

5) Statement on increased extramedullary hematopoiesis cannot rely on increased spleen weight only. E.g. spleen histology must be studied for it. Moreover, are there differences is the number of other than erythroid blood cells?

6) The results in Figure 6I and 6N are somewhat conflicting. Enhanced lactate clearance does not suggest impaired lactate tolerance. Please, comment.

7) Moreover, the data presented in Figure 6L and 6M is not very convincing and rather irrelevant for the topic. Why would HIF-2α loss in CB cells impair glucose tolerance?

8) The authors should prepare a conclusion illustration.

*Reviewer #3:*

The authors present experimental evidence indicating that HIF-2α expression is required for the survival (in newborn) and/or proliferation (in adult) of carotid body (CB) glomus cells, which in turn are required for CB-mediated responses to hypoxia.

1) Since Cre expression is directed by TH promoter, it would be helpful to analyze another marker to demonstrate loss of glomus cells in the KO mice.

2) Are neurons in the rostral ventrolateral medulla or nucleus tractus solitarius that receive input from the CB affected by HIF-2α^KO^ in TH-expressing cells?

3) Subsection “Sympathoadrenal Hif-2α loss blocks carotid body glomus cell development”, last paragraph – the authors show an effect of HIF-2α^KO^ on glomus cell survival, not differentiation.

4) Subsection “Deficient acclimatization to chronic hypoxia in mice with CBs loss”, first paragraph – do the authors really believe that GFAP expression is sufficient to identify stem cells?

5) Subsection “Deficient acclimatization to chronic hypoxia in mice with CBs loss”, second paragraph – histology data is needed here.

6) Subsection “Deficient acclimatization to chronic hypoxia in mice with CBs loss”, last paragraph – authors' conclusion is puzzling. The changes in the lungs seem to explained by greater hypoxemia and polycythemia in the KO mice, rather than any change intrinsic to the lungs.

7) Subsection “Deficient acclimatization to chronic hypoxia in mice with CBs loss”, first paragraph – statistical analysis is needed here.

8) Please delete "the time to exhaustion trended lower in mutants".

9) Subsection “Adaptive responses to exercise and high glucose are affected in mice with CB dysfunction”, first paragraph – lactate findings are curious since the KO mice are more hypoxemic and therefore should produce more lactate; in addition, lactate is thought to cause muscle exhaustion. Some discussion of these results would be helpful.

10) Please delete "TH expression trended lower in both CB and SCG TH^+^ cells…"

11) What is the mechanism by which HIF-2α regulates mTorC1 activity?

12) Many studies of cultured cells have demonstrated that mTorC1 activity is inhibited by hypoxia. Please discuss.

13) Do the HIF-2α^KO^ mice show normal responses to hypercarbia, which are also mediated by the CB?

14) What is the phenotype of mice with loss of both HIF-1α and HIF-2α in TH-expressing cells?

---

## [Author Response]

Reviewer #1:

[…] i) As the authors outline (Discussion, first paragraph), the original descriptions of constitutive HIF-α inactivation phenotypes included the association of HIF-2α inactivation with failure of sympatho-adrenal development. The authors then state that the current findings argue against a 'global role for the (HIF-2α) gene in development of catecholamine tissues'. The most obvious difference between this and the earlier reports is the use of TH-IRES-Cre 'driver' versus constitutive inactivation. What do the authors know of the timing of Cre activation using this strain of mouse? Is it possible that the TH promoter used in the driver-strain only becomes active during later stages of sympatho-adrenal development, and that the current (carotid body – restricted) phenotype reflects this i.e. a more extensive phenotype might have been generated if the deletion of HIF-2α had occurred earlier in the sympatho-adrenal lineage? Further discussion of this is required to set the work in the context of that earlier literature.

We agree with the reviewer that deletion of HIF-2α at earlier stages of sympatho-adrenal development might result in a more extensive phenotype as this may potentially affect migration, proliferation and/or differentiation of neural crest-derived sympatho-adrenal precursors, which would impact in developmental catecholamine homeostasis as reported (Tian, Hammer, Matsumoto, Russell, and McKnight, 1998).

To address this issue, we have used a knock-in TH-IRES-Cre mouse strain that uses the endogenous TH (tyrosine hydroxylase) promoter to drive Cre expression. Gene deletion thus occurs in all TH-expressing cells. CB primordium is first seen at E13 in the mouse embryo (Kameda, 2005; Kameda, Nishimaki, Takeichi, and Chisaka, 2002; Kondo, 1975). At that embryonic stage, when the CB is still not developed, there are abundant numbers of TH^+^ cells both in the superior cervical ganglion (SCG) and in the adrenal medulla (Huber et al., 2002; Kameda, 2005). However, our findings show that these tissues are essentially unaffected by HIF-2α deletion. We have amended the sentence ‘specific role of HIF-2α in the development of the CB glomus cells, and argues against a global role for the gene in development of catecholaminergic tissues’ to specify ‘late development of catecholaminergic tissues’. Additionally, we have now clarified this issue in the Discussion (second paragraph).

ii) The authors describe extensive (and interesting) physiological differences between wild type mice and the TH-HIF-2α^KO^ animals. Some of these differences occur in normoxic animals and some (e.g. pulmonary vascular responses) are seen only in response to hypoxia. If possible it would be important to distinguish those effects which are directly due to abnormal carotid body function (i.e. altered neurogenic responses) from those are secondary to increased hypoxia. For example, it is stated that 'pulmonary adaption to chronic hypoxia is highly reliant on a normally functioning CB'. For the general reader it would be important to make it clear that such a connection may well be indirect, (due to increased hypoxia) rather implying a direct role of the carotid body. Along similar lines, what is known of the oxygen saturation of the TH-HIF-2α^KO^ animals in normoxia? Are they always fully saturated – or do they manifest desaturation at any stage? Even transient desaturation could have important effects on the parameters that are being described. Ideally data on this, or at least clear discussion as to whether this has or has not been excluded, is important.

We agree with the reviewer that some of the phenotypes observed are directly related to the abnormal CB function and others are a consequence of a failed CB-dependent acclimatization to hypoxia (e.g., exaggerated erythrocytosis and pulmonary vascular remodeling). The sentence was misleading and has been amended (subsection “Deficient acclimatization to chronic hypoxia in mice with CBs loss”, last paragraph). Also, as suggested by the reviewing editor and reviewer 2, we have added a summary figure (Figure 8) that we hope will clarify this point.

The reviewer has suggested monitoring oxygen saturation of the TH-HIF-2α^KO^ mice, as fluctuations in normoxia might be important for the physiological parameters studied. Although we cannot monitor O_2_ saturation for 24 hours, due to stringent UK legal and ethical concerns that limit repeated or prolonged procedures in experimental animals, we have now measured peripheral oxygen saturation in anaesthetized TH-HIF-2α^KO^ mice breathing 21% O_2_ for a period of 20 minutes. This data has been added to Figure 3 (Figure 3H) and described in the text (subsection “Impaired ventilatory response and whole-body metabolic activity in Th-Epas1^KO^ mice exposed to hypoxia”, second paragraph).

iii) In the first paragraph of the subsection “CB development and proliferation in response to hypoxia require mTORC1 activation” the authors argue that hypoxia is an mTORC1 activator. Greater clarity on both the background literature and interpretation of the current work is required – particularly as hypoxia can also be an mTORC1 inhibitor due to metabolic effects. Specifically it would be helpful if the authors could discuss briefly what connections between HIF-2α and mTORC1 have been established previously and what (direct or indirect connections) they are proposing on the basis of the current work.

We appreciate the reviewer’s suggestion. We have now clarified the contrasting effects of hypoxia in regulating cell proliferation (subsection “CB development and proliferation in response to hypoxia require mTORC1 activation”, first paragraph) and altered the sentence (last paragraph). Also, we have added some examples from the literature citing evidence for HIFα regulation of mTORC1 activation (Discussion, seventh paragraph).

With that said, there are not many examples in the literature that link HIF-2α and mTORC1 activity under physiological conditions. Elorza, et al., reported in 2012 that HIF-2α transcriptionally regulates the expression of the amino acid transporter *Slc7a5,* and that this in turn activates mTORC1 activity (Elorza et al., 2012). We have tried to reproduce this observation a number of times now, unfortunately without success. Analysis of this amino acid transporter in the CB of TH-HIF-2α^KO^ mice by immunohistochemistry was not able to be done in a convincing manner. The small size of the CB and the fact that the CB glomus cells were dying during development hampered further molecular characterization in our hands.

iv) Although early work apparently implicated HIF-1α as a key factor in carotid body regulation, the current work builds on a body on more recent data indicating the importance of HIF-2α rather than HIF-1α in carotid body physiology. In the Introduction the authors state 'mouse models hemizygous for HIFs-α or where HIF inactivation was induced in adults showed contrasting effects on CB function without in either case affecting CB morphogenesis'. In fact the paper by Hodson et al. examined both hemizygous animals and biallelic inactivation in the adult. It reported that in both settings HIF-2α (not HIF-1α) inactivation had major effects on functional responses to sustained hypoxia and that the proliferative (i.e. morphological) response of the carotid bodies to sustained hypoxia was entirely ablated after HIF-2α but not HIF-1α^KO^. These findings are closely relevant to (and consistent with) the current work and should be described more clearly in the introduction. Notwithstanding this, I do consider that the current work is an important advance as outlined above.

We thank the reviewer for this observation. We meant ‘without affecting CB *developmental* morphogenesis’. This sentence has now been modified (Introduction, fourth paragraph). We have also expanded the Introduction to point out the findings reported in the recent work by Hodson, et al..

v) Subsection “Sympathoadrenal Hif-2α loss blocks carotid body glomus cell development”, last paragraph. I am not sure that the greater number of TUNEL+ cells in HIF-2α^KO^ animals indicates an essential function in differentiation (as opposed to survival). Could the authors clarify or modify?

This sentence has now been modified (subsection “Sympathoadrenal Epas1 loss blocks carotid body glomus cell development”, last paragraph).

vi) Can the authors comment on potential reasons why the hypoxic HIF-2α^KO^ animals suffered haemorrhages?

The lack of CB-driven compensatory response to hypoxia in the mutant mice clearly leads to a more severe hypoxemia. As a result, aberrant polycythemia and pulmonary vasoconstriction occur; these can lead to failure of the pulmonary cardiovascular circuit. A closer look to TH-HIF-2α^KO^ mice after hypoxia exposure revealed ascites and congestive hepatomegaly, which indicates a potential right ventricular failure. Internal haemorrhages in the gut were observed only in mice found dead in the chamber. We have added this description to the manuscript (subsection “Deficient acclimatization to chronic hypoxia in mice with CBs loss”, second paragraph).

vii) Figure 1—figure supplement 1A please clarify designation of upper and lower panels i.e. TH-Cre = (activated tDTomato).

This has now been modified in the figure 1—figure supplement 1 and the figure legend accordingly.

viii) Figure 3—figure supplement 1 – is there a greater response to CO_2_ in the TH-HID-2α adrenal medulla or is this appearance due to different scales – could the authors comment?

We are sure the reviewer appreciates that electrophysiological experiments are extremely complex, and that sometimes it is difficult to show in a single recording visually representative responses of all the conditions studied. Amperometry is a technique that allows us to measure both the charge and the frequency of secretory events upon stimulation. Therefore, a recording showing smaller (less cargo) but more numerous spikes will have similar secretion rates than another with larger (more cargo) but lower frequency events. Averaged secretory rates are shown in Figure 3—figure supplement 2D for all the conditions studied.

Reviewer #2:

[…] The manuscript contains a very large amount of high quality data from demanding experimental set ups and is well written. However, it lacks coherence and a clear conclusion. The amount of data provided is overwhelming and some reduction should be considered to make the manuscript more coherent. The manuscript would certainly benefit from a conclusive figure (schematic).I have several concerns that warrant my enthusiasm for the publication of the manuscript.1) What is the deletion efficiency of HIF1α and HIF2α in the corresponding CB KO cells? In adult TH-HIF2α^KO^s it may not be measurable but at P0 or E18.5 CB cells are still available for HIF2α^KO^s.

We agree that deletion efficiency should be quantified in CB glomus cells of TH-HIF-1α^KO^ and TH-HIF-2α^KO^ mice. Unfortunately, the tiny size of the CB at those stages hamper a clean dissection to some extent. Our genetic approach shows a clear activation of the tdTomato reporter in the CB upon TH-IRESCre recombination (Figure 1—figure supplement 1A). Since we observed a noticeable phenotype in the CB TH^+^ cells we felt that it was important to determine the deletion efficiency in TH^+^ tissues without a phenotype, such as the SCG and the AM (Figure1—figure supplement 1).

Additionally, we do have extensive experience with this mouse strain (THIRES-Cre) and we have previously seen highly tissue-specific phenotypes in CB glomus cells (Diaz-Castro, Pintado, Garcia-Flores, Lopez-Barneo, and Piruat, 2012; Fernandez-Aguera et al., 2015; Macias, Fernandez-Aguera, Bonilla-Henao, and Lopez-Barneo, 2014).

2) The conclusion that mTORC1 activation is required for CB expansion in a HIF2α-dependent manner cannot be drawn from the experiments performed. In order to draw such a conclusion for example the rapamycin experiment should be repeated with TH-HIF2α^KO^s. Regarding the mechanism, did rapamycin induce apoptosis of CB cells?

In response to this concern, we have now performed experiments in TH-HIF2α^KO^ mice injected with rapamycin. New data have been added to Figure 7E-I and commented on in the text (subsection “CB development and proliferation in response to hypoxia require mTORC1 activation”, fourth paragraph).

We agree with the reviewer that testing whether or not rapamycin induces apoptosis would be a great addition to the mechanism. We have tried on a number of occasions to perform this experiment using a similar approach to that used for embryonic tissue (i.e., TUNEL staining on carotid bifurcation slices). However, in these cases, we either see no TUNEL+ cells in the CB of adult mice (with/without rapamycin).

We have also performed histological quantification of CB volume and TH^+^ cell numbers of wild type mice with or without rapamycin, and did not see significant changes. These data have been added to figure 7—figure supplement 1G and H and described in the manuscript (subsection “CB development and proliferation in response to hypoxia require mTORC1 activation”, fourth paragraph).

3) As both, VO_2_ and VCO_2_ change after hypoxia (Figure 3J and K) – please show RER.

This data have been included in Figure 3K and N and a comment added to the text (subsection “Impaired ventilatory response and whole-body metabolic activity in Th-Epas1^KO^ mice exposed to hypoxia”, third paragraph).

4) In the Angiotensin II-induced hypertension model the difference in TH-HIF2α^KO^ BP is only seen at 7 day (not at 14 or 21 days). What might be the reasons for this?

Angiotensin II increases arterial pressure by a number of actions, including vasoconstriction, sympathetic stimulation and renal actions via releasing aldosterone and antidiuretic hormone. Eventually, chronic infusion of Ang II also induces vascular and renal damage (Casare et al., 2016; Griffin et al., 1991). The modest but significant effects on baseline blood pressure observed in our TH-HIF-2α^KO^ mouse model are likely due to a decreased sympathetic tone in the absence of CB stimulation. In the Ang II-induced hypertension model, this absence of CB-driven sympathetic stimulation can only partially counteract the effects of Ang II in the short term and delay the emergence of hypertension. However, ultimately, Ang II actions that are independent of the sympathoadrenal system, along with classically described hypertension-induced vascular and renal injuries, do give rise to an uncontrolled hypertension in the model.

5) Statement on increased extramedullary hematopoiesis cannot rely on increased spleen weight only. E.g. spleen histology must be studied for it. Moreover, are there differences is the number of other than erythroid blood cells?

We thank the reviewer for this observation. We have now performed histology of the spleen and this data has been added to Figure 4G. Also, we have quantified other blood cells after hypoxia treatment and the new data are included in Figure 4—figure supplement 1E-H. This data are also commented on (subsection “Deficient acclimatization to chronic hypoxia in mice with CBs loss”, second paragraph).

6) The results in Figure 6I and 6N are somewhat conflicting. Enhanced lactate clearance does not suggest impaired lactate tolerance. Please, comment.

Blood lactate concentration is one of the most commonly measured parameters during clinical exercise testing. A rise in blood lactate levels upon incremental exertion is well documented in the literature (Owles, 1930). THHIF-2α^KO^ mice show poorer exercise performance, consistent with the VO_2_ and VCO_2_ rates described here. This decreased metabolic activity during exertion leads to a decrease in lactate accumulation at exhaustion (Figure 6I).

It has been reported in one recent, high profile publication that CB glomus cells are able to monitor plasma lactate levels (Chang, Ortega, Riegler, Madison, and Krasnow, 2015). Thus, we thought it important to test the validity of that observation by our challenge of TH-HIF-2α^KO^ mice with an intraperitoneal lactate bolus, in order to follow its clearance over time (Figure 6N).

7) Moreover, the data presented in Figure 6L and 6M is not very convincing and rather irrelevant for the topic. Why would HIF-2α loss in CB cells impair glucose tolerance?

We respectfully disagree with the reviewer. CB has been reported to be sensitive to glucose levels through a number of different experimental approaches (Gao et al., 2014; Garcia-Fernandez, Ortega-Saenz, Pardal, and Lopez-Barneo, 2003; Pardal and Lopez-Barneo, 2002). Given the importance of the sympathoadrenal system in glucose homeostasis (Rodriguez-Diaz et al., 2011), we thought that it would be important to document whether CB loss indirectly regulates glucose levels.

8) The authors should prepare a conclusion illustration.

This has now been added to the manuscript (Figure 8).

Reviewer #3:The authors present experimental evidence indicating that HIF-2α expression is required for the survival (in newborn) and/or proliferation (in adult) of carotid body (CB) glomus cells, which in turn are required for CB-mediated responses to hypoxia.1) Since Cre expression is directed by TH promoter, it would be helpful to analyze another marker to demonstrate loss of glomus cells in the KO mice.

We agree with the reviewer that this is a valid concern. We have performed immunohistochemistry to detect serotonin (5-HT) and TH on CB bifurcations at E18.5 embryo stage (see Author response image 1). Serotonin expression was restricted to CB TH^+^ cells and vice versa.

TH is considered by workers in the field to be a reliable marker for catecholaminergic cells, and CB glomus cells in particular. The TH-IRES-Cre mouse strain is a transgenic knock-in which does not appear to affect the expression of the endogenous *Th* gene (Lindeberg et al., 2004). Our histological analysis shows that TH-HIF-2α^KO^ mice have normal TH expression in the adult SCG and embryonic CB glomus cells. It should be noted as well that we see that adult TH-HIF-1α^KO^ mice do not show defects in the CB at any detectable level. This is itself a control that Cre expression does not interfere with *Th* gene. This is consistent with previous work of ours using this model in CB glomus cells (Fernandez-Aguera et al., 2015).

**Author response image 1. respfig1:** CB glomus cells identification by serotonin expression. Double serotonin (5-HT, red) and TH (green) immunofluorescence on carotid bifurcations from HIF-2α^WT^ and TH-HIF-2α^KO^ mice at E18.5. Scale bars: 100 µm.

2) Are neurons in the rostral ventrolateral medulla or nucleus tractus solitarius that receive input from the CB affected by HIF-2α^KO^ in TH-expressing cells?

We agree that the lack of stimulation of afferent nerve fibers by the CB might trigger the re-setting or dysfunction of neurons of the respiratory center. This is an interesting question that certainly has to be explored in the future. Our plethysmograh experiments, however, suggest that this is not occuring here, as TH-HIF-2α^KO^ mice show normal respiratory parameters at 21% O_2_. Additionally, and as suggested by the reviewer (see comment 13 below), we have now performed plethysmograph recordings in response to hypercapnia (5% CO_2_) and observed normal responses (Figure 3B and C). Although the CB is sensitive to hypercapnia, up to 80% of the response is due to central chemoreceptors of the respiratory center (Guyenet, Stornetta, and Bayliss, 2010; Heeringa, Berkenbosch, de Goede, and Olievier, 1979). Therefore, it is very unlikely that the lack of response to hypoxia in TH-HIF-2α^KO^ mice is due to defects in the central chemoreceptors. Nonetheless, we have performed a basic histological analysis of the nucleus of the solitary tract as shown in Author response image 2. We did not observe evident alterations in the neurons stained with the pan-neuronal marker NeuN.

**Author response image 2. respfig2:** Histological analysis of the nucleus of the solitary tract. NeuN immunohistochemistry of brain slices from HIF-2α^WT^ andTH-HIF-2α^KO^ mice. Left picture was obtained from Allen brain atlas (http://mouse.brain-map.org/static/atlas). Nucleus of the solitary tract is highlighted in purple. Dash line depicts the area corresponding with the nucleus of the solitary tract. Scale bars: 500 µm.

3) Subsection “Sympathoadrenal Hif-2α loss blocks carotid body glomus cell development”, last paragraph – the authors show an effect of HIF-2α^KO^ on glomus cell survival, not differentiation.

This has now been corrected.

4) Subsection “Deficient acclimatization to chronic hypoxia in mice with CBs loss”, first paragraph – do the authors really believe that GFAP expression is sufficient to identify stem cells?

We agree with the reviewer’s concern here. We note that we referred to a particular set of stem cells, the type II or sustentacular cells of the carotid bodies. These cells (GFAP+) have been identified as resident adult stem cells responsible for the CB proliferation and growth in response to hypoxia giving rise to both glomus and endothelial cells in vitro and in vivo (Annese, NavarroGuerrero, Rodriguez-Prieto, and Pardal, 2017; Navarro-Guerrero et al., 2016; Pardal, Ortega-Saenz, Duran, and Lopez-Barneo, 2007; Platero-Luengo et al., 2014).

5) Subsection “Deficient acclimatization to chronic hypoxia in mice with CBs loss”, second paragraph – histology data is needed here.

We agree. Histology data has now been added in Figure 4.

6) Subsection “Deficient acclimatization to chronic hypoxia in mice with CBs loss”, last paragraph – authors' conclusion is puzzling. The changes in the lungs seem to explained by greater hypoxemia and polycythemia in the KO mice, rather than any change intrinsic to the lungs.

We have now modified this statement.

7) Subsection “Deficient acclimatization to chronic hypoxia in mice with CBs loss”, first paragraph – statistical analysis is needed here.

Statistical analysis was performed, this sentence has now been amended.

8) Please delete "the time to exhaustion trended lower in mutants".

This sentence has been deleted.

9) Subsection “Adaptive responses to exercise and high glucose are affected in mice with CB dysfunction”, first paragraph – lactate findings are curious since the KO mice are more hypoxemic and therefore should produce more lactate; in addition, lactate is thought to cause muscle exhaustion. Some discussion of these results would be helpful.

The treadmill experiments were performed in ambient oxygenation. A raise in blood lactate levels upon incremental exertion is classically described (Owles, 1930). This increase in plasma lactate levels is to a great extent a result of muscle metabolism. Exercise in rodents has a mild impact on arterial pO_2_ (Dempsey and Wagner, 1999). Therefore, we believe the results seen after running of TH-HIF-2α^KO^ mice is an indirect effect of CB loss on sympathetic tone, one that ultimately impacts on exercise performance, rather than an exercise-induced hypoxemia.

10) Please delete "TH expression trended lower in both CB and SCG TH^+^ cells…"

This sentence has now been deleted.

11) What is the mechanism by which HIF-2α regulates mTorC1 activity?

As noted above, there are not many examples in the literature that link HIF-2α and mTORC1 activity under physiological conditions. Elorza, et al., reported in 2012 that HIF-2α transcriptionally regulates the expression of the amino acid transporter *Slc7a5* and that this in turn activates mTORC1 activity (Elorza et al., 2012). We have tried to reproduce this observation a number of times now, unfortunately without success. Analysis of this amino acid transporter in the CB of TH-HIF-2α^KO^ mice by immunohistochemistry was not able to be done in a convincing manner. The small size of the CB and the fact that the CB glomus cells were dying during development hampered further molecular characterization in our hands.

12) Many studies of cultured cells have demonstrated that mTorC1 activity is inhibited by hypoxia. Please discuss.

Examples of these studies have now been added in the Discussion (seventh paragraph).

13) Do the HIF-2α^KO^ mice show normal responses to hypercarbia, which are also mediated by the CB?

We thank the reviewer for this suggestion. We have now performed plethysmograph experiments in response to 5% CO_2_. These new data have been included in Figure 3B and C, and commented on in the text (subsection “Impaired ventilatory response and whole-body metabolic activity in Th-Epas1^KO^ mice exposed to hypoxia”, first paragraph).

14) What is the phenotype of mice with loss of both HIF-1α and HIF-2α in TH-expressing cells?

This would of course be an interesting animal model to study, however, the generation of conditional double knockouts for Hif-1α and Hif-2α is genetically very time consuming, and unfortunately beyond the time frame given for this revision. We will certainly attempt to do this experiment in the future.